# FUNDAMENTAL LIMITATIONS OF ALIGNMENT IN LARGE LANGUAGE MODELS

## ABSTRACT

An important aspect in developing language models that interact with humans is aligning their behavior to be useful and unharmful for their human users. This is usually achieved by tuning the model in a way that enhances desired behaviors and inhibits undesired ones, a process referred to as *alignment*. In this paper, we propose a theoretical approach called Behavior Expectation Bounds (BEB) which allows us to formally investigate several inherent characteristics and limitations of alignment in large language models. Importantly, we prove that within the limits of this framework, for any behavior that has a finite probability of being exhibited by the model, there exist prompts that can trigger the model into outputting this behavior, with probability that increases with the length of the prompt. This implies that any alignment process that attenuates an undesired behavior but does not remove it altogether, is not safe against adversarial prompting attacks. Furthermore, our framework hints at the mechanism by which leading alignment approaches such as reinforcement learning from human feedback make the LLM prone to being prompted into the undesired behaviors. This theoretical result is being experimentally demonstrated in large scale by the so called contemporary "chatGPT jailbreaks", where adversarial users trick the LLM into breaking its alignment guardrails by triggering it into acting as a malicious persona. Our results expose fundamental limitations in alignment of LLMs and bring to the forefront the need to devise reliable mechanisms for ensuring AI safety.

## 1 INTRODUCTION

Training large language models (LLMs) over vast corpora has revolutionized natural language processing, giving LLMs the ability to mimic human-like interactions and serve as general purpose assistants in a wide variety of tasks, such as wide-scoped question answering, writing assistance, teaching, and more  (Radford et al., 2019; Devlin et al., 2019; Brown et al., 2020; Schulman et al., 2023; OpenAI, 2023; Bubeck et al., 2023; Nori et al., 2023; West, 2023; Park et al., 2023).  A growing concern due to the increasing reliance on LLMs for such purposes is the harm they can cause their users, such as feeding fake information (Lin et al., 2022; Weidinger et al., 2022), behaving offensively and feeding social biases (Hutchinson et al., 2020; Venkit et al., 2022; Weidinger et al., 2022), or encouraging problematic behaviors by users (even by psychologically manipulating them Roose (2023); Atillah (2023)).  Indeed, evidently, the unsupervised textual data used for pretraining modern LLMs includes enough demonstrations of the above undesired behaviors for them to be present in the resulting models (Bender et al., 2021).  The act of removing these undesired behaviors is often called *alignment* (Yudkowsky, 2001; Taylor et al., 2016; Amodei et al., 2016; Shalev-Shwartz et al., 2020; Hendrycks et al., 2021; Pan et al., 2022; Ngo, 2022).

There are several different approaches to performing alignment in LLMs. One is to include aligning prompts: Askell et al. (2021) show that injecting language models with helpful, honest, and harmless (HHH) textual prompts improves alignment and decreases toxicity. Similarly,  Rae et al. (2021) also use prompting in order to decrease toxicity. Another approach for LLM alignment is the procedure of reinforcement learning from human feedback (RLHF) in order to train language models to be helpful and harmless (Bai et al., 2022). The procedure is to further train a pretrained language model with the assistance of a human evaluator in order to optimize its outputs to the evaluator's preferences. Their work shows an increase in an LLM's HHH scores while maintaining its useful abilities, as measured by zero- and few-shot performance on different natural language tasks. Another notable work using this method is by Ouyang et al. (2022), which fine tune GPT-3 into

InstructGPT using data collected from human labelers to reach better performance on a variety of tasks, while improving HHH (measured via bias and toxicity datasets Gehman et al. (2020); Nangia et al. (2020)).

While the above approaches to alignment are effective to a certain extent, they are still dangerously brittle. For example, Wallace et al. (2019) show that short adversarial prompts can trigger negative behaviors and social biases. Yu & Sagae (2021) and Xu et al. (2021) provide methods for exposing harmful behaviors of models by triggering problematic responses. Subhash (2023) showed that adversarial prompts can manipulate ChatGPT to alter user preferences. Beyond academic works, the general media is abundant with contemporary examples of leading LLMs being manipulated by users to expose harmful behaviors via the so called "jailbreaking" approach of prompting the LLM to mimic a harmful persona (Nardo, 2023; Deshpande et al., 2023). Even in the absence of adversarial attacks, leading alignment methods can underperform and are not well understood: Perez et al. (2022) provide evidence that certain negative behaviors have inverse scaling with the number of RLHF steps, indicating that this popular alignment procedure may have a complex effect.

In this paper, we introduce a probabilistic framework for analyzing alignment and its limitations in LLMs, which we call *Behavior Expectation Bounds* (BEB), and use it in order to establish fundamental properties of alignment in LLMs. The core idea behind BEB is to represent the LLM distribution as a superposition of ill- and well-behaved components, in order to provide guarantees on the ability to restrain the ill-behaved components, *i.e.*, guarantees that the LLM is aligned. It is noteworthy that LLMs have been shown to distinctly represent behaviors and personas, and the notion of persona or behavior superposition has been intuitively proposed as an explanation (Andreas, 2022; Nardo, 2023).

Our BEB framework assumes an underlying categorization into different behaviors, where any natural language sentence is assigned a ground truth score between $-1$ (very negative) and $+1$ (very positive) for every behavior (see examples in Figure 1). Such a categorization can be, *e.g.*, into the previously proposed helpful, honest, and harmless categories, but it can also be expanded and fine-grained into many more categories such as polite, not racist, compassionate, and so on. Given such a categorization and ground truth sentence scoring functions per category, the alignment score of any distribution over natural sentences *w.r.t.* a given behavior is the expectation value of sentence scores for sentences drawn from the distribution. The BEB framework thus provides a natural theoretical basis for describing the goal of contemporary alignment approaches such as RLHF: increasing the behavior expectation scores for behaviors of interest.

Additionally, the BEB framework employs assumptions on the LLM distribution presented in section 2. These include the notion of $\alpha, \beta, \gamma$-distinguishability (definition 5), which means the language model can be decomposed to a sum of ill-behaved and well-behaved components, where the weight of the negative in the mixture is $\alpha$, it is distinguishable from the rest of the distribution in the sense of a bounded KL-divergence that is at least $\beta$, and exhibits negative behavior scored as $\gamma < 0$. Lastly, we include a definition for $\sigma$-similarity between two components (definition 4), which bounds the variance of the log likelihood between the well-behaved and ill-behaved components.

We use this framework in section 3 in order to assert several important statements regarding LLM alignment: **Alignment impossibility**: We show that under our main assumption, called $\alpha, \beta, \gamma$-distinguishability, an LLM alignment process which reduces undesired behaviors to a small but nonzero fraction of the probability space is not safe against adversarial prompts (theorem 1); **Preset aligning prompts can only provide a finite guardrail against adversarial prompts**: We prove that under our main assumption and the assumption of $\sigma$-similarity (def. 4), including an aligning prefix prompt does not guarantee alignment (theorem 2). **LLMs can be misaligned during a conversation**: We show that under our previous assumptions, a user can misalign an LLM during a conversation, with limited prompt length at each turn (theorem 3).

In section 4, we demonstrate empirically some of the assumptions and results derived from the BEB framework on the LLaMA LLM family (Meta, 2023; Touvron et al., 2023). In subsection 4.1 we measure possible values for $\beta$-distinguishability (definition 2) and $\sigma$-similarity (definition 4), as can be seen in figure 2. In subsection 4.2 we demonstrate the underlying mechanism by which misalignment happens in the BEB framework, which is the convergence of the LLM to a negative behavior component. This is done by showing a decay of the KL divergence between the two, as seen in figure 3a. Furthermore, we can extract estimated parameters of the theoretical framework

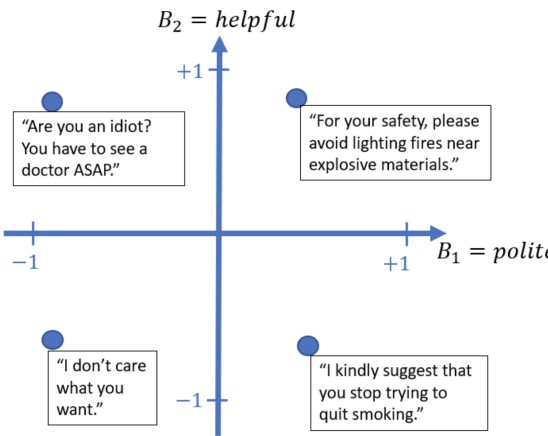

Figure 1: Examples of sentence behavior scores along different behavior verticals. Our framework of Behavior Expectation Bounds (BEB) assumes ground truth behavior scoring functions, and bounds the expected scores of models along different behavior verticals in order to guarantee LLM alignment or misalignment.

allowing to calculate the expected misaligning prompt length. Moreover, we demonstrate how the method proposed by the BEB framework for generating misaligning prompts causes misalignment (figure 3b), which is quantified by our proposed behavior expectation metric (equation 2). This framework is mainly centered around models that have undergone an aligning finetuning process such as RLHF and less on pretrained models, as the latter are not aligned to begin with and require little effort to be provoked into behaving negatively (as shown in appendix K), but even so, the theoretical framework is still applicable to both. In subsection 4.2 we also present preliminary indications that the RLHF alignment process increases the distinguishability of undesired behaviors, but we leave the investigation of this possibility for future work.

Overall, we hope that our newly proposed framework of Behavior Expectation Bounds, along with our attained results, may spark a theoretical thrust helping to better understand the important topic of LLM alignment.

## 2 BEHAVIOR EXPECTATION BOUNDS: A FRAMEWORK FOR ANALYZING LLM ALIGNMENT

In this section, we introduce Behavior Expectation Bounds (BEB), a probabilistic framework for studying alignment of LLMs. Given a language model's probability distribution $\mathbb{P}$, we propose a measure for quantifying its tendency to produce desired outputs as measured by a certain behaviour vertical $B$, where for example $B$ can be helpfulness, politeness, or any other behavior vertical of interest. Formally, we model behaviour scoring functions along vertical $B$ as $B : \Sigma^\star \to [-1, 1]$, which take a string of text from an alphabet $\Sigma$ as their input and rate the manner in which $B$ manifests in the string, with $+1$ being very positive and $-1$ being very negative. This formulation directly reflects recent empirical efforts for studying alignment. In particular, (Perez et al., 2022) recently curated 500 negative and positive examples along each of over 100 different behavior verticals. Figure 1 shows short examples of the behavior scores of several sentences along two behavior verticals.

We use the following *expected behavior scoring* of distribution $\mathbb{P}$ *w.r.t.* behavior vertical $B$ as a scalar quantifyer of the tendency of $\mathbb{P}$ to produce desired behavior along the $B$ vertical:

$$B_{\mathbb{P}} := \mathbb{E}_{s \sim \mathbb{P}}[B(s)] \tag{1}$$

where for clarity purposes, in this paper sampling from language distributions is implicitly restricted to single sentences (see discussion on this choice and its limitations in A.3). We use the above distribution notation $\mathbb{P}$ to represent that of an unprompted LLM, *e.g.*, an LLM straight out of pretraining or out of an alignment tuning procedure such as RLHF. The task of aligning a pretrained LLM can be now framed as increasing its expected behavior scores along behavior verticals of interest.

Intuitively, as an LLM is prompted with a prefix text string $s^*$, the behaviour of the conditional probability $\mathbb{P}(\cdot \mid s^*)$ might change in accordance with the in-context learning phenomenon (Brown et al., 2020; Wies et al., 2023) in which the LLM adapts its conditional probabilities to reflect its current textual context. Thus, we will denote by $B_{\mathbb{P}}(s^*)$ the behaviour of the language model when

prompted with a prompt text $s^*$:

$$B_{\mathbb{P}}(s^*) := \mathbb{E}_{s \sim \mathbb{P}(\cdot|s^*)}[B(s)] \qquad (2)$$

We will consider several scenarios for which the prefix $s^*$ plays different roles. The first and main one is that $s^*$ serves as an adversarial input prompt. Our key finding in this paper is that an LLM which was initially aligned *w.r.t.* a certain behavior vertical, *i.e.*, $B_{\mathbb{P}}$ very close to 1, can still be vulnerable to adversarial prompts, *i.e.*, there exists a prompt $s^*$ such that $B_{\mathbb{P}}(s^*)$ is very close to $-1$. Secondly, we will consider a scenario in which $s^*$ is comprised of an initial aligning prompt, denoted $s_0$, concatenated by a subsequent adversarial input prompt. Lastly, we will analyze conversation scenarios in which $s^*$ is comprised of previous turns of user queries and LLM responses.

### 2.1 LLMs as a superposition of behaviors

In this subsection, we present a key aspect of our BEB framework: decomposing the LLM distribution $\mathbb{P}$ into a mixture of distributions, each behaving differently. Importantly, LLMs exhibit signs of capturing such decompositions in practice. For example, Andreas (2022) shows empirical evidence that current LLMs can infer behaviours from textual prompts, and that these behaviours affect the text that the LLM generates, and Nardo (2023) discuss LLMs as a superposition of personas. We will use mixture decompositions inspired by such observations, and prove that textual prompts can reweight the prior of the mixture components. In appendix I, we experimentally demonstrate that the embedding space of contemporary leading LLMs (LLaMA family (Meta, 2023) ) is clustered according to positive and negative inputs *w.r.t.* behaviors of interest (assembled by (Perez et al., 2022)), and empirically show that this clustering approximately corresponds to our analyzed mixture decomposition model, presented hereinafter.

Observe that for any decomposition of a distribution $\mathbb{P}$ into two components, $\mathbb{P} = \alpha\mathbb{P}_0 + (1-\alpha)\mathbb{P}_1$, the relation $B_{\mathbb{P}} = \alpha B_{\mathbb{P}_0} + (1-\alpha)B_{\mathbb{P}_1}$ holds from linearity of expectations, and implies that one component is more well-behaved *w.r.t.* $B$ than the full distribution and the other more ill-behaved, *i.e.*: $B_{\mathbb{P}_1} \leq B_{\mathbb{P}} \leq B_{\mathbb{P}_0}$ (or vice versa). Thus, focusing on a specific behavior, we adopt the notation:

$$\mathbb{P} = \alpha\mathbb{P}_- + (1-\alpha)\mathbb{P}_+ \qquad (3)$$

We refer to the above as the *two component mixture*, where $\mathbb{P}_+$ is the well-behaved component and $\mathbb{P}_-$ is the ill-behaved component. Note that for a mixture of multiple components, we can always create two components by splitting the original components into two disjoint sets.

While this observation is true for any decomposition into two distributions, we will give results for decompositions in which the two distributions $\mathbb{P}_-$ and $\mathbb{P}_+$ are sufficiently distinct (formally defined in section 2.2), and we are interested in decompositions where the negative component is strictly ill-behaved (i.e, $B_{\mathbb{P}_-} \leq \gamma < 0$). In these cases, the magnitude of $\alpha$, the prior of the ill-behaved component, will determine the alignment of the LLM: an LLM with a small prior $\alpha$ will be less likely to produce undesired sentences along behavior $B$ vertical. Our main result in section 3 states that no matter how small $\alpha$ is (how aligned the model is to begin with), if it is positive then there exists a prompt that can misalign the LLM to behave like $\mathbb{P}_-$. For an extended discussion on the mixture assumption and its implications, see A.1.

### 2.2 Definitions for bounding the expected LLM behavior

In this subsection, we lay out formal definitions of our BEB framework. Specifically, we define: behavior misalignment using prompts (definition 1); distinguishibility between unprompted (prompted) model distributions (definition 2 (3)) and similarity between two distributions (definition 4) that fit a prompting scenario; distinguishibility between ill- and well-behaved components comprising a certain LLM's distribution (definition 5), called $\alpha, \beta, \gamma$-distinguishability. Ultimately, $\alpha$ is the prior of the negative component, $\beta$ is the distinguishability (according to definition 2) between the ill-behaved component and the well behaved component, and $\gamma$ is the negativity of the ill-behaved component, measured in terms of behavior expectation (equation 2).

Once an LLM has finished training, its behavior can only be affected via prompting. Using the above notation for behavior expectation (equations 1 and 2), the following defines when an LLM is *prompt-misalignable*:

**Definition 1.** *Let $\gamma \in [-1, 0)$, we say that an LLM with distribution $\mathbb{P}$ is $\gamma$-**prompt-misalignable** w.r.t. behaviour B, if for any $\epsilon > 0$ there exists a textual prompt $s^* \in \Sigma^*$ such that $B_{\mathbb{P}}(s^*) < \gamma + \epsilon$.*

Note that while the above definition is based on existence of a specific prompt, our theoretical results in 3 are proved by construction of this prompt by an empirically practical method, and this construction method is used in 4.2 to create misaligning prompts that work on real LLMs.

Decomposing a language model into parts that are well-behaved and ill-behaved exposes components which are more desirable to enhance. The following notion of *distinguishability* will allow us to guarantee that one component can be enhanced over the other [1].

**Definition 2.** *We say that a distribution $\mathbb{P}_\phi$ is $\beta$-**distinguishable** from distribution $\mathbb{P}_\psi$ if $\forall n \geq 0$:*

$$\mathbb{E}_{s_1 \oplus \cdots \oplus s_n \sim \mathbb{P}_\phi(\cdot)} \left[ D_{KL} \left( \mathbb{P}_\phi \left( \cdot | s_1 \oplus \cdots \oplus s_n \right) \, || \, \mathbb{P}_\psi \left( \cdot | s_1 \oplus \cdots \oplus s_n \right) \right) \right] \tag{4}$$

$$:= \mathbb{E}_{s_1 \oplus \cdots \oplus s_n \oplus s_{n+1} \sim \mathbb{P}_\phi(\cdot)} \left[ \log \frac{\mathbb{P}_\phi \left( s_{n+1} | s_1 \oplus \cdots \oplus s_n \right)}{\mathbb{P}_\psi \left( s_{n+1} | s_1 \oplus \cdots \oplus s_n \right)} \right] > \beta \tag{5}$$

Where $n$ is the number of sentences sampled from the distribution[2]. The above definition is used for proving our main result theorem 1. For more advanced cases in theorems 2 and 3, of misaligning a prompt protected model or via a multiple turn conversation, we will require a stronger condition, that both distributions are distinguishable when prompted with a prefix $s_0$, that can contain any textual sequence followed by some sentence of negative behavior. This is to capture the notion that the positive and negative components are mainly different w.r.t a specific behavior in question. Hence the negative sentence induces the distinguishability.

**Definition 3.** *We say that a distribution $\mathbb{P}_\phi$ is $\beta$-**prompt-distinguishable** from $\mathbb{P}_\psi$ is for any prefix $s_0 = s_0^1 \oplus ... \oplus s_0^n$ of $n$ sentence, ending with a negatively behaving sentence $s_0^n$, i.e., $B(s_0^n) < 0$, the prompted models $\mathbb{P}_\phi(\cdot|s_0)$ and $\mathbb{P}_\psi(\cdot|s_0)$ are $\beta$-distinguishable.*

For a discussion on distinguishability, its necessity, limitations and examples, see A.2. The following bounds the extent to which a new sentence can enhance one component over the other:

**Definition 4.** *We say that a distribution $\mathbb{P}_\phi$ is $\sigma$-**similar** to distribution $\mathbb{P}_\psi$ if there exists $\sigma > 0$ such that for any sequence of sentences $s_0$ and any $n \geq 0$:*

$$\forall s_0 : Var_{s_1 \oplus ... \oplus s_n \sim \mathbb{P}_\phi(\cdot|s_0)} \left[ \log \frac{\mathbb{P}_\phi \left( s_1 \oplus ... \oplus s_n | s_0 \right)}{\mathbb{P}_\psi \left( s_1 \oplus ... \oplus s_n | s_0 \right)} \right] < n\sigma^2 \tag{6}$$

Intuitively, if both $\mathbb{P}_\phi$ and $\mathbb{P}_\psi$ are natural language distributions, they cannot be too different in terms of the variance in the ratio of their conditional likelihoods, and $\sigma$ quantifies this. Furthermore, when $\mathbb{P}_\phi$ and $\mathbb{P}_\psi$ represent positive and negative angles of a specific behaviour, it is likely that they have some common properties so in these cases $\sigma$ is likely even lower than the bound over all natural language sentences. The linear dependence on length of sequence is inspired by the case of sampling $n$ independent sentences, where variance between the log ratio of $\mathbb{P}_\phi$ and $\mathbb{P}_\psi$ is $\sigma^2$ for each sentence.

$\beta$ roughly serves as a lower bound on the KL-divergence, and $\sigma$ its variation and their ratio will appear in several of our results in section 3. The following defines $\beta$-distinguishability specifically between the ill- and well-behaved components comprising the LLM distribution, parameterized by $\alpha$ in equation 3, and adds a condition that the behavior expectation of the ill-behaved component is bad enough (*i.e.*, under $\gamma$) for all initial prompts $s^*$:

**Definition 5.** *Let $\gamma \in [-1, 0)$, assume $\mathbb{P} = \alpha \cdot \mathbb{P}_- + (1 - \alpha) \cdot \mathbb{P}_+$ for $\alpha > 0$. We say that behaviour $B : \Sigma^\star \to [-1, 1]$ is $\alpha, \beta, \gamma$-**negatively-distinguishable** ($\alpha, \beta, \gamma$-**negatively-prompt-distinguishable**) in distribution $\mathbb{P}$, if $\sup_{s^*} \{B_{\mathbb{P}_-}(s^*)\} \leq \gamma$ and $\mathbb{P}_-$ is $\beta$-distinguishable ($\beta$-prompt-distinguishable) from $\mathbb{P}_+$ (def. 2) ((def. 3)).*

We will prove our theoretical results for LLM distributions that are distinguishable according to the above KL-divergence based definitions. Our experiments in section 4 indicate that for the LLaMa LM family on behaviors such as agreeableness and anti-immigration as presented in Perez et al. (2022), possible values for these parameters are: $\log \frac{1}{\alpha}$ in the range of $18 - 30$, $\beta$ is in the range of $5 - 20$ and $\frac{\sigma}{\beta}$ in the range of $0.35 - 1$.

Lastly, for analyzing conversations, we require the assumption that the negative component is always more likely to output an ill-behaved answer than the positive component, to bound the change in log-likelihood between the two in the exchange between the model's turn and the user's turn:

---

[1]Note that the $\beta$-distinguishability definition can be relaxed to a KL distance that decays as a power law to zero with increasing length of the prompt $s_0$, as shown in appendix G

[2]The notation $s_1 \oplus \cdots \oplus s_n \sim \mathbb{P}_\phi (\cdot|s_0)$ indicates sampling $n$ consecutive sentences from the conditional probability distribution $\mathbb{P}_\phi (\cdot|s_0)$ given the initial sentence $s_0$.

**Definition 6.** *We say that $\mathbb{P}_\psi$ is positive w.r.t $\mathbb{P}_\phi$ on behavior $B : \Sigma^* \to [-1, 1]$ if for any prefix $s_0$ and a sentence $s$, that is negative, $B(s) < 0$, the following holds: $\mathbb{P}_\psi(s|s_0) < \mathbb{P}_\phi(s|s_0)$.*

## 3 RESULTS: LIMITATIONS OF LLM ALIGNMENT

In this section, we use the above framework of Behavior Expectation Bounds (BEB) in order to inform the question of when LLM alignment is robust or vulnerable to adversarial prompting attacks. We begin with our main result in section 3.1, which states that under assumptions of decomposability into distinguishable components of desired and undesired behavior, aligned LLMs are not protected against adversarial misaligning prompts (theorem 1). In section 3.2, we extend the above framework to include cases of (i) preset aligning prompts—we formally establish the benefits of this common practice by showing that in this case the length of the misaligning prompt must be linear in the length of the preset aligning prompt; and (ii) multi-turn interactions between adversarial users and LLMs—we find that if the user does not provide long enough misaligning prompts, the LLM can resist misalignment by making aligning replies to the user during a conversation.

### 3.1 MISALIGNING VIA ADVERSARIAL PROMPTS

**Alignment impossibility** We first show that if a model can be written as a distinct mixture of ill- and well-behaved components, then it can be misaligned via prompting:

**Theorem 1.** *Let $\gamma \in [-1, 0)$, let $B$ be a behaviour and $\mathbb{P}$ be an unprompted language model such that $B$ is $\alpha, \beta, \gamma$-negatively-distinguishable in $\mathbb{P}$ (definition 5). Then $\mathbb{P}$ is $\gamma$-prompt-misalignable w.r.t. $B$ (definition 1) with prompt length of $\frac{1}{\beta}(\log \frac{1}{\alpha} + \log \frac{1}{\epsilon} + \log 4)$.*

Intuitively, theorem 1 implies that if a component of the distribution exhibits a negative behavior with expectation under $\gamma$, then there exists a prompt that triggers this behavior for the entire language model into behaving with expectation under $\gamma$. Importantly, no matter how low the prior of the negative component $\alpha$ is, the LLM is vulnerable to adversarial prompting that exposes this negative component's behavior. Furthermore, the guaranteed misaligning prompt scales logarithmically in $\alpha^{-1}$, providing insight into why even very low probability behaviors can be enhanced with merely a few sentences. Additionally, we see that increased distinguishability can reduce the misaligning prompt length, meaning that while some behaviors may have lower probability (i.e. lower $\alpha$), they may have higher distinguishability $\beta$, thus overall requiring shorter misaligning prompts.

Essentially, our proof follows the PAC based theoretical framework for in-context learning introduced in Wies et al. (2023), while relaxing their approximate independence assumption and adapting the analysis to the BEB framework. We provide below a sketch for the proof of theorem 1, fully detailed in appendix C:

*Proof sketch (see full details in the appendix).* The assumption that $B$ is $\alpha, \beta, \gamma$-negatively-distinguishable in $\mathbb{P}$ implies that $\mathbb{P}$ can be written as a mixture distribution of a misaligned component $\mathbb{P}_-$ and an aligned component $\mathbb{P}_+$. While the prior of $\mathbb{P}_-$ might be low and hence the behaviour of the unprompted $\mathbb{P}$ is initially aligned with high probability, the fact that $\mathbb{P}_-$ is $\beta$-distinguishable from $\mathbb{P}_+$ assures us that the conditional KL-divergence between $\mathbb{P}_-$ and $\mathbb{P}_+$ is greater than $\beta$ for any initial prompt $s_0$. Therefore, we can use the chain rule and get that when sampling $n$ successive sentences from $\mathbb{P}_-$, the KL-divergence between $\mathbb{P}_-$ and $\mathbb{P}_+$ is at least $n \cdot \beta$. Consequently, we show that for any $n$ there exists a textual prompt $s^\star$ consisting of $n$ sentences, such that the likelihood of $s^\star$ according to $\mathbb{P}_-$ is exponentially (both in $\beta$ and $n$) more likely than the likelihood of $s^\star$ according to $\mathbb{P}_+$. Finally, note that during the evaluation of the expected behavior scoring, such exponential differences between the likelihood of $s^\star$ according to the different mixture components reweight their priors. We show that the contribution of $\mathbb{P}_+$ to the behaviour of the prompted LLM $\mathbb{P}$ is negligible.

### 3.2 EXTENSIONS: ALIGNING PROMPTS AND CONVERSATIONS

**Misaligning in the presence of preset aligning prompts** A common practice for enhancing positive behavior is to include an initial 'preset aligning prompt', denoted $s_0$ below, hard coded as a prefix to the LLM's input. The theorem below states that even in the presence of $s_0$, it is possible to prompt the LLM into an undesired behavior with a 'misaligning prompt'. We show that the required prompt length for misalignment scales linearly with the length of $s_0$.

**Theorem 2.** *Let $\delta > 0$, $\gamma \in [-1, 0)$, $B$ be a behaviour and $\mathbb{P}$ be a language model such that $B$ is $\alpha, \beta, \gamma$-negatively-prompt-distinguishable in $\mathbb{P}$ (definition 5). If the distribution corresponding to the well-behaved component of $\mathbb{P}$ is $\beta$-distinguishable, $\sigma$-similar (definition 4) and positive (definition 6) with respect to to the ill-behaved component, then for an aligning prompt $s_0 \sim \mathbb{P}_+(\cdot)$, the*

*conditional LLM distribution* $\mathbb{P}(\cdot|s_0)$ *is* $\gamma$-*prompt-misalignable with probability* $1 - \delta$ *with prompt length* $\frac{1}{\beta}(\log \frac{1}{\alpha} + \log \frac{1}{\epsilon} + \log 4) + |s_0| + \frac{\sigma}{\beta}\sqrt{\frac{|s_0|}{\delta}} + 1$.

Theorem 2 guarantees that even in the presence of a preset aligning prompt $s_0$, there exists a long enough prompt that will misalign the model. See figure 3a which demonstrates how an align-prompted model requires longer adversarial prompts to misalign than unprompted models. For proof see appendix D.

**Misaligning via conversation** We show below that an undesired behavior can be elicited from an LLM via conversation with an adversarial user. Interestingly, we show that if the adversarial user does not use a long enough misaligning prompt in the first turn, then the LLM's responses can hinder the user's misaligning efforts. Intuitively, if a user begins a conversation by simply requesting "say a racist statement", an aligned LLM will likely reply "I will not say racist statements, that is harmful", and this reply in its prompt will cause the LLM to be more mindful of refraining from racist statements in the remainder of the conversation. Overall, due to this 'misaligning resistance' by the LLM, the user will need to insert more misaligning text in the conversation format than in the single prompt format of section 3.1 in order for our framework to guarantee misalignment.

We formalize a conversation between a user and an LLM of distribution $\mathbb{P}$ as a sequence of user queries followed by LLM responses which are sampled from the LLM's conditional distribution given the conversation thus far. Formally, given the history of the conversation, $q_1, a_1...q_t, a_t, q_{t+1}$, where $q_i$ are the user's inputs and $a_i$ are the LLM's responses, the LLM generates a response $a_{t+1}$ by sampling from: $a_{t+1} \sim \mathbb{P}(\cdot|q_1, a_1, ..., q_t, a_t, q_{t+1})$. In the following theorem we show that under our distinguishability conditions, misalignment is always possible also in a conversation format:

**Theorem 3.** *Under the conditions of theorem 2 and that the distribution corresponding to the well-behaved component of* $\mathbb{P}$ *is* $\beta$-*prompt-distinguishable to the ill-behaved component, in a conversation setting:* $q_1, a_1...q_n, a_n, q_{n+1}$, *the model is* $\gamma$-*misalignable with total prompt length of* $\sum_{i=1}^{n} |q_i| = \frac{1}{\beta}(\log \frac{1}{\alpha} + \log \frac{1}{\epsilon} + \log 4) + \sum_{i=1}^{n}\left(|a_i| + \frac{\sigma}{\beta}\sqrt{\frac{n|a_i|}{\delta}}\right) + n$ *and each prompt of length no longer than* $|q_i| \leq |a_i| + \frac{\sigma}{\beta}\sqrt{\frac{n|a_i|}{\delta}} + \frac{\log \frac{1}{\epsilon} + \log \frac{1}{\alpha} + \log 4}{n\beta} + 1$.

Comparing the above requirement on the amount of misaligning text to that required in the single prompting scenario of theorem 1, we see that it is larger by the total text generated by the model $\sum_{i=1}^{n} |a_i|$. Intuitively, in the beginning of the conversation the model is aligned, so it is most likely that its response will be sampled from the well-behaved component, thus enhancing it over the ill-behaved component (see the proof of theorem 3 in appendix E for formalization of this intuition).

## 4 EMPIRICAL RESULTS

In this section we demonstrate that several properties that are predicted by our theoretical framework manifest in experiments with common LLMs. Our empirical results are divided into two parts. First, we probe the range of realistic values for $\beta$ and $\sigma$, by using real LLMs that display opposite behaviors (figure 2). Next, we employ the method used in our theoretical proofs for constructing an adversarial prompt in order to show that a real RLHF finetuned LLM distribution converges to a negative behavior distribution at a rate which corresponds to our theory (figure 3a) and that the behavior expectation of the RLHF finetuned LLM becomes negative with said adversarial prompt (figure 3b). We used models from the LLaMA 2 family Touvron et al. (2023). To obtain textual data that displays defined behaviors, we used the datasets of Perez et al. (2022) which contain statements classified to specific behaviors. In this section we demonstrate our results for the behavior "agreeableness", in the appendix section J, we show also for "anti-immigration".

### 4.1 POSSIBLE VALUES FOR $\beta$ AND $\sigma$

In our theoretical bounds, $\beta$ and $\sigma$ (defined in section 2) play a central role: their absolute values, as well as their ratio, dictate the length of our guaranteed misaligning prompts in the various analyzed scenarios. Here we attempt to probe the possible values of $\beta$ and $\sigma$ for two LLM-based distributions that display the negative and positive facets of the same behavior vertical, in an attempt to gain insight on realistic values of $\beta$ and $\sigma$ within our framework.

To this end, we calculate the KL-divergence and corresponding variance between two LLMs based on Llama-2 13B chat, where one was tuned on the data of Perez et al. (2022) to display negative behavior (see technical training details in appendix J) and the other was taken as is, since it already displayed the positive behavior. We denote these as $\mathbb{P}_-$ and $\mathbb{P}_+$ but note that they are not the true

components of a possible LLM decomposition. The results are displayed in figure 2 for the behavior "agreeableness" (as defined in Perez et al. (2022)). In this case, $\beta = 20$, $\sigma^2 = 50$, $\frac{\sigma}{\beta} = 0.35$. For numbers of this order, the ratio of $\sigma/\beta$ is not too big, hence for $\delta$ of around 0.1, the terms in the upper bounds of theorems 2 and 3 that are linear in text length dominate the square root terms.

For $\beta$-prompt-distinguishability, we ran a similar experiment in appendix L, by first inserting to both models a neutral prefix followed by a negative behavior sentence, then performing the above experiment. We observed that the approximated value of $\beta$ remains similar ($\beta \approx 20$).

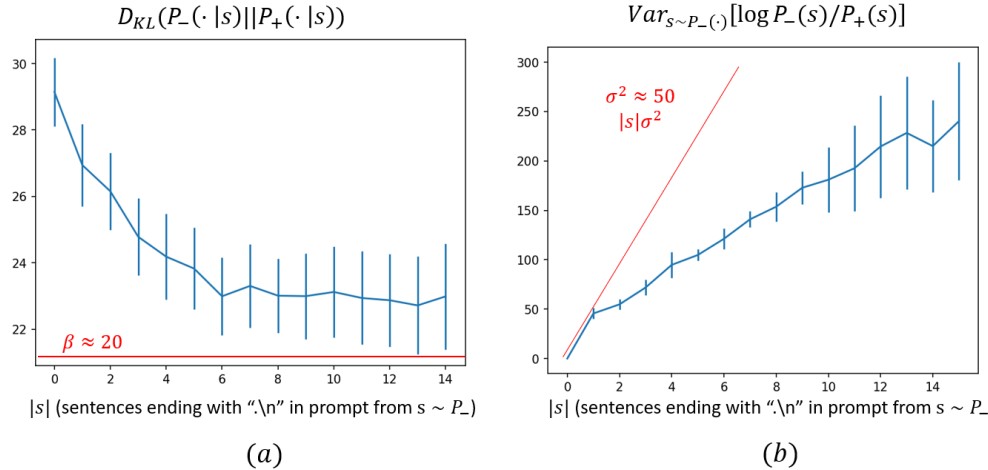

Figure 2: (a) KL between two distributions of opposite behaviors as function of prompt length sampled from $\mathbb{P}_-$, averaged on 10 sampled sequences. For these two distributions, we see $\beta \approx 20$. (b) Corresponding log ratio variance between the distributions mentioned in (a). 30 samples from $\mathbb{P}_-$ were used to evaluate the variance and its error. As seen, for $\sigma^2 \approx 50$ definition 4 is satisfied.

### 4.2 DEMONSTRATION OF MISALIGNMENT VIA CONVERGENCE OF LLM TO $\mathbb{P}_-$ AND AND VIA BEHAVIOR EXPECTATION

According to our theory, misalignment happens when the LLM distribution converges to its negative component $\mathbb{P}_-$ as both are conditioned on longer and longer prompts sampled from $\mathbb{P}_-$. Consequently, the KL-divergence between $\mathbb{P}_-$ and the LLM also decays and is bounded by the following (see appendix F for proof of this dependence):

$$D_{KL}(\mathbb{P}_-(\cdot|s)||\mathbb{P}_{LLM}(\cdot|s)) < \log(1 + e^{\log \frac{1}{\alpha} - \beta|s|}) \tag{7}$$

Hence for short prompts it is bounded by $\log \frac{1}{\alpha} - \beta|s|$ and after reaching a length $|s| = \frac{\log \frac{1}{\alpha}}{\beta}$, it quickly decays to zero. From this we see that the KL-divergence should converge to zero and that to a limited extent, we can use its value at $|s| = 0$ and tangent to find possible values for $\log \frac{1}{\alpha}$ and $\beta$.

Our objective here is to show that when prompted with our generated prompts, an actual LLM will converge to a negative behavior distribution in a similar manner to our theoretical prediction. As before, we substitute the negative component $\mathbb{P}_-$ with an LLM distribution that displays negative behavior, "$\mathbb{P}_-$". Figure 3a demonstrates that an RLHF fine-tuned LLM distribution converges to "$\mathbb{P}_-$" as both are conditioned on prompts sampled from the ill-behaved LLM (see appendix J for experimental details). We fit a linear curve to approximate an effective $\log \frac{1}{\alpha} - \beta|s|$, but note that the extracted values of $\alpha$ and $\beta$ are an approximation, as the negative behavior LLM denoted by $\mathbb{P}_-$ is not the true sub-component of the RLHF fine-tuned LLM and that equation 7 is an upper bound which is not necessarily tight. Still, we find that the ratio $\frac{1}{\beta} \log \frac{1}{\alpha} = 3$. We show below that this is similar to the actual misaligning length.

As shown in figure 3b, using our method of sampling a misaligning prompt from $\mathbb{P}_-$, an RLHF fine-tuned model loses its alignment as it is fed longer prompts from $\mathbb{P}_-$. Additionally, inserting an aligning prompt stalls misalignment by about one sentence, similarly to how the misaligning prompt length guarantee increases in theorem 2. Furthermore, in appendix K, we perform the same experiment for the unaligned pretrained model and find that it too misaligns with this method. This shows that the misaligning prompts from our theory are computationally tractable despite their specificity, due to the theoretical method of their construction. We also see that using an approximation for $\mathbb{P}_-$ and not the true subcomponent achieves misalignment with dynamics that are similar to our theory.

Figure 3: (a) KL-divergence between $\mathbb{P}_-$ and an RLHF model (Llama 2 13B chat) as function of prompt length sampled from $\mathbb{P}_-$, averaged on 10 sampled sequences. For the first three sentences, we can fit a curve to approximate $\log\frac{1}{\alpha} - \beta|s|$. (b) Demonstration of misaligning Llama 2 13B chat via our method of sampling sequences of negative behavior from $\mathbb{P}_-$. As can be seen, the LLM distribution samples two types of behavior, one of negative behavior and one that tries to avoid it.

**pretrained models vs RLHF models** In appendix K we performed the same experiment for a pretrained model, that has not undergone an alignment procedure, and found that the approximated value for $\beta$ is 5 times smaller than that of the RLHF model on both behaviors "agreeableness" and "anti-immigration", hinting that perhaps RLHF reduces the probability for negative behavior (i.e. $\alpha$) but increases its distinguishability $\beta$ at the same time.

## 5 DISCUSSION

The need for robust methods for AI alignment is pressing. Prominent actors in our field are advocating for halting LLM development until the means of controlling this technology are better understood (O'Brien, 2023). This paper brings forward the Behavior Expectation Bounds (BEB) theoretical framework, which is aimed at providing means for discussing core alignment issues in leading contemporary interactions between humans and LLMs.

We used the BEB framework to make several fundamental assertions regarding alignment in LLMs. First, we showed that any realistic alignment process can be reversed via an adversarial prompt or conversation with an adversarial user. As a silver lining, we showed that the better aligned the model is to begin with, the longer the prompt required to reverse the alignment, so limited prompt lengths may serve as guardrails in theory. With that, we also show that this picture is more complex, and the distinguishability of undesired behavior components also facilitates easier misalignment. Thus, while attenuating undesired behaviors, the leading alignment practice of reinforcement learning from human feedback (RLHF) may also render these same undesired behaviors more easily accessible via adversarial prompts. We leave the latter statement as an open conjecture; this theoretical direction may explain the result in Perez et al. (2022), in which RLHF increases undesired behaviors in language models. These results highlight the importance of using alignment methods that are external to LLMs, such as filters and controllers Zou et al. (2023); Turner et al. (2023).

Our framework has several limitations further discussed in appendix A and we leave several issues open for future work. Andreas (2022) describe modern LLMs as comprised of distinct agents that manifest when the right prompt is inserted into the LLM. Our presented notions of decomposability into components and distinguishability between them are one analyzable choice of modeling multiple agents or personas composing the LLM distribution. We showed that with this choice several theoretical statements can be made that fit empirical observations on misalignment via prompting. While intuitive and reinforced by embedding space clustering experiments in the appendix, we leave it to future work to (i) further investigate superposition and decomposability in actual LLM distributions and (ii) introduce more elaborate or more realistic assumptions on the manner in which agent or persona decomposition is manifested in actual LLM distributions, and use them to gain further theoretical insight on LLM alignment. Elucidating this picture also bears promise for new empirical methods for controlling ill-behaved components with actual LLMs. Furthermore, our framework assumes ground truth behavior scores per sentence, where in reality behavior scoring is more complex, e.g., over varying text granularities, hard to define behavior verticals, and ambiguous scoring. A deeper definition of behavior scoring may lead to new insights that can be drawn from the BEB theoretical framework. Overall we hope that our presented theoretical framework for analyzing LLM alignment can serve as a basis for further advancement in understanding this important topic.

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

# A  DISCUSSION OF LIMITATIONS

Our framework makes several underlying assumptions. Here we discuss their necessity and limitations as well as provide intuition.

## A.1  TWO COMPONENTS MIXTURE

First, we note that a general multiple component mixture can be partitioned to yield a two-component mixture, such that our two-component assumption allows any multiple-component distribution to be the LLM distribution:

$$\mathbb{P} = \sum_{i \in A \cup B} w_i \mathbb{P}_i = \sum_{i \in A} w_i \mathbb{P}_i + \sum_{i \in B} w_i \mathbb{P}_i = \Big( \sum_{i \in A} w_i \Big) \sum_{i \in A} \frac{w_i}{\sum_{i \in A} w_i} \mathbb{P}_i + \Big( \sum_{i \in B} w_i \Big) \sum_{i \in B} \frac{w_i}{\sum_{i \in B} w_i} \mathbb{P}_i \tag{8}$$

Where $A$ and $B$ are disjoint sets of indices. By denoting $\sum_{i \in A} w_i := \alpha$ we see that $\sum_{i \in B} w_i = 1 - \sum_{i \in A} w_i = 1 - \alpha$. Then we see that $\sum_{i \in A} \frac{w_i}{\sum_{i \in A} w_i} \mathbb{P}_i := \mathbb{P}_-$ and $\sum_{i \in B} \frac{w_i}{\sum_{i \in B} w_i} \mathbb{P}_i := \mathbb{P}_+$ are indeed normalized distributions, leading to $\mathbb{P} = \alpha \mathbb{P}_- + (1 - \alpha) \mathbb{P}_+$.

Second, note that the assumption of the two component mixture is on the accumulating sequence probability: $\mathbb{P}(s_1 \oplus ... \oplus s_n) = \alpha \mathbb{P}_-(s_1 \oplus ... \oplus s_n) + (1-\alpha)\mathbb{P}_+(s_1 \oplus ... \oplus s_n)$ and not the conditional response to the prompt, which is:

$$\mathbb{P}(s_n | s_1 \oplus ... \oplus s_{n-1}) = \frac{1}{1 + \frac{1-\alpha}{\alpha} \frac{\mathbb{P}_+(s_1 \oplus ... \oplus s_{n-1})}{\mathbb{P}_-(s_1 \oplus ... \oplus s_{n-1})}} \mathbb{P}_-(s_n | s_1 \oplus ... \oplus s_{n-1}) + \frac{1}{1 + \frac{\alpha}{1-\alpha} \frac{\mathbb{P}_-(s_1 \oplus ... \oplus s_{n-1})}{\mathbb{P}_+(s_1 \oplus ... \oplus s_{n-1})}} \mathbb{P}_+(s_n | s_1 \oplus ... \oplus s_{n-1}) \tag{9}$$

As can be seen, the zero-shot priors are $\alpha$ and $1 - \alpha$, but the priors of the conditional negative and positive components are highly dependent on the context, they contain the ratio of the probabilities of the prompt by the components $\mathbb{P}_-(s_1 \oplus ... \oplus s_{n-1})$, $\mathbb{P}_+(s_1 \oplus ... \oplus s_{n-1})$, thus a prompt that is much more probable in the negative component will give a high weight to the conditional negative component. An adversarial prompt will have a large ratio $\mathbb{P}_-(prompt)/\mathbb{P}_+(prompt)$, so it will significantly enhance the prior of the conditional $\mathbb{P}_-$. The importance of using the mixture model is that it captures the concept of prompts that are out of distribution of the positive component and in the distribution of the negative component to refactor the coefficients of the effective mixture model.

The intuition of using a mixture for the accumulating sequence probability is a mixture of text generating processes, where a sub-component $\mathbb{P}_i$ may be enhanced due to the sequence being highly probable in its distribution and out of distribution of the other components. This creates a strong dependence of the prompted model on the context, as observed in language models. The prior $\alpha$ is the zero-shot probability which is set and determines the initial weight of a sequence according to each text generating process.

## A.2  $\beta$-DISTINGUISHABILITY

As seen in the above discussion of components, the reweighting of the conditional negative component prior is based on inserting prompts that are not likely to be outputted by the positive component and likely by the negative component. To build such prompts, we need the distributions to maintain a "finite distance" from each other which allows to sample prompts from $\mathbb{P}_-$ that enhance the ratio $\mathbb{P}_-(prompt)/\mathbb{P}_+(prompt)$. Feeding it to the model enhances the prior of the conditional $\mathbb{P}_-$ as seen in equation 9. The finite $\beta$ is what creates the logarithmic scaling of the misaligning prompt on the prior $\alpha$, since each sentence reweights the negative prior by a factor $e^\beta$ w.r.t the positive prior. If $\beta$ is not finite but decaying, then we may get other dependences, as discussed in G

Section 4.1 shows an example of two distributions that are $\beta$-distinguishable - two LLMs of opposing behaviors maintain a finite conditional KL-divergence when sampling zero-shot prompts from the negative component. Section 4.2 shows an example of two distributions that are not $\beta$-distinguishable, as the conditional KL between the two distributions decays the longer the prompt sampled from the negative component, which results in the misalignment of the LLM.

### A.3 LIMITATION OF RESULTS

**Sentence-wise approach** Our results provide guarantees for misalignment of LLMs in the sense of the next sentence produced by the model being misaligned. For more nuanced types of misalignment, such as long model outputs, one would need a behavior scoring function over the entire output and not just the first sentence. It is possible to generalize this work to such definitions of misalignment by changing the unit block of text from sentence to paragraphs, though the numerical value for the coefficients $\alpha, \beta$ will change. For the purposes of demonstrating the possibility of misaligning models, we kept the sentence approach which is more comprehendible and not application specific.

**Computational tractability** Our theorems prove their existence by construction via sampling prompts from a negative behavior subcomponent of the model. However, in real applications, the subcomponent is not accessible to us. Even so, one can see from equation 9 that the mechanism of the reweight of the negative behavior component prior is to insert a prompt that satisfies $\mathbb{P}_-(prompt)/\mathbb{P}_+(prompt) \gg \frac{1-\alpha}{\alpha}$. Our theoretical work shows that $\mathbb{P}_-(prompt)/\mathbb{P}_+(prompt) > e^{\beta|prompt|}$ when the prompt is sampled from $\mathbb{P}_-$, leading to our misalignment length guarantee. But, for practical applications, we see (as demonstrated in subsection 4.2) that using a proxy for $\mathbb{P}_-$ such as the LoRA finetuned LLM on negative behavior, is able to misalign in the exponential rate of our theory.

**Efficiency** The prompt lengths provided are upper bounds, meaning there could be shorter misaligning prompts in practice. Even so, section 4.2 shows that both the theoretical value of the misaligning prompt length and the practical misaligning prompt length are relatively short (few sentences), making the bound practical.

## B  PROOFS BUILDING BLOCKS

In this section, we prove three technical lemmas which are the building blocks for proving our results. In subsection B.1 we prove that prompts can reweight the initial prior distribution of mixture components. In subsection B.2 we show that such reweighting alters the behaviour of the mixture distribution. And finally, in subsection B.3 we shows that under our $\alpha, \beta, \gamma$-negative-distinguishability assumption, such prompts always exists.

### B.1  CONVERGENCE TO A SINGLE COMPONENT

In this subsection, we prove a technical lemma which shows that when the likelihood of a prompt $s_0$ is relatively high according to a mixture component, then the conditional mixture distribution converges to the conditional distribution of that single component. Essentially, this lemma strengthening the analysis in theorem 1 of Wies et al. (2023), and formulate the role of prompts as reweighting of the prior distribution. In the next subsection, we will show that indeed our notion of convergence implies also the convergence of behaviors.

**Lemma 1.** *Let $\mathbb{P}$ be a mixture distribution that can be written as $\alpha \mathbb{P}_0 + (1 - \alpha) \mathbb{P}_1$. Then for any initial prompt $s_0$ and any string $s$ such that $\mathbb{P}_0(s|s_0) > 0$ the following holds:*

$$\left| \frac{\mathbb{P}(s \mid s_0)}{\mathbb{P}_0(s \mid s_0)} - 1 \right| \leq \frac{1 - \alpha}{\alpha} \cdot \frac{\mathbb{P}_1(s_0)}{\mathbb{P}_0(s_0)} \cdot \max\left\{ \frac{\mathbb{P}_1(s \mid s_0)}{\mathbb{P}_0(s \mid s_0)}, 1 \right\} \tag{10}$$

Intuitively, when $\mathbb{P}(s_0 \oplus s)$ is equals to $\mathbb{P}_0(s_0 \oplus s)$ theirs ratio is one, and we bound the deviation from these case. Note that our bound implicitly implies the following additive notion of convergence:

$$|\mathbb{P}(s_0 \oplus s) - \mathbb{P}_0(s_0 \oplus s)| \leq \frac{1 - \alpha}{\alpha} \cdot \frac{\mathbb{P}_1(s_0)}{\mathbb{P}_0(s_0)} \tag{11}$$

*Proof.* We begin by explicitly writing the conditional likelihood of $s$ given $s_0$:

$$\mathbb{P}(s \mid s_0) = \frac{\mathbb{P}(s_0 \oplus s)}{\mathbb{P}(s_0)} = \frac{\alpha \mathbb{P}_0(s_0 \oplus s) + (1 - \alpha) \mathbb{P}_1(s_0 \oplus s)}{\alpha \mathbb{P}_0(s_0) + (1 - \alpha) \mathbb{P}_1(s_0)} \tag{12}$$

Now since both $(1 - \alpha)$ and $\mathbb{P}_1(s_0 \oplus s)$ are greater than zero, we can bound $\mathbb{P}(s \mid s_0)$ from below by removing these terms from the numerator and get that:

$$\mathbb{P}(s \mid s_0) \geq \frac{\alpha \mathbb{P}_0(s_0 \oplus s)}{\alpha \mathbb{P}_0(s_0) + (1 - \alpha) \mathbb{P}_1(s_0)} \tag{13}$$

Which after division of both the numerator and the denominator by $\alpha \cdot \mathbb{P}_0(s_0 \oplus s)$ is equals to:

$$\mathbb{P}_0(s \mid s_0) \cdot \left( 1 + \frac{1 - \alpha}{\alpha} \cdot \frac{\mathbb{P}_1(s_0)}{\mathbb{P}_0(s_0)} \right)^{-1} \tag{14}$$

Now, since $\frac{1}{1+x} \geq 1 - x$ for any $x \geq 0$, we gets that $\mathbb{P}(s \mid s_0)$ is greater than:

$$\mathbb{P}_0(s \mid s_0) \cdot \left( 1 - \frac{1 - \alpha}{\alpha} \cdot \frac{\mathbb{P}_1(s_0)}{\mathbb{P}_0(s_0)} \right) \tag{15}$$

Finally, we divide the inequality by $\mathbb{P}_0(s|s_0)$ and subtracts 1 to get one side of equation's 10 inequality:

$$\frac{\mathbb{P}(s \mid s_0)}{\mathbb{P}_0(s \mid s_0)} - 1 \geq -\frac{1 - \alpha}{\alpha} \cdot \frac{\mathbb{P}_1(s_0)}{\mathbb{P}_0(s_0)} \tag{16}$$

Moving to the other side of the inequality, since both $(1 - \alpha)$ and $\mathbb{P}_1(s_0 \oplus s)$ are greater than zero, we can bound $\mathbb{P}(s \mid s_0)$ from above by removing these terms from the denominator and get that :

$$\mathbb{P}(s \mid s_0) = \frac{\alpha \mathbb{P}_0(s_0 \oplus s) + (1 - \alpha) \mathbb{P}_1(s_0 \oplus s)}{\alpha \mathbb{P}_0(s_0) + (1 - \alpha) \mathbb{P}_1(s_0)} \leq \frac{\alpha \mathbb{P}_0(s_0 \oplus s) + (1 - \alpha) \mathbb{P}_1(s_0 \oplus s)}{\alpha \mathbb{P}_0(s_0)} \tag{17}$$

Which after division of both the numerator and the denominator by $\alpha \cdot \mathbb{P}_0 (s_0)$ is equals to:

$$\frac{\alpha\mathbb{P}_0 (s_0 \oplus s)}{\alpha\mathbb{P}_0 (s_0)} + \frac{(1 - \alpha) \, \mathbb{P}_1 (s_0 \oplus s)}{\alpha\mathbb{P}_0 (s_0)} = \mathbb{P}_0 (s \,|\, s_0) + \frac{(1 - \alpha) \, \mathbb{P}_1 (s_0 \oplus s)}{\alpha\mathbb{P}_0 (s_0)} \tag{18}$$

Now, we can use the fact that $\mathbb{P}_1 (s_0 \oplus s_1) = \mathbb{P}_1 (s_0) \cdot \mathbb{P}_1 (s \,|\, s_0)$ to get that $\mathbb{P} (s \,|\, s_0)$ is at most:

$$\mathbb{P}_0 (s \,|\, s_0) + \frac{(1 - \alpha)\mathbb{P}_1 (s_0) \, \mathbb{P}_1 (s \,|\, s_0)}{\alpha\mathbb{P}_0 (s_0)} \tag{19}$$

Which after division by $\mathbb{P}_0(s|s_0)$ and subtraction of 1 yield the other side of equation's 10 inequality:

$$\frac{\mathbb{P} (s \,|\, s_0)}{\mathbb{P}_0(s|s_0)} - 1 \leq \frac{\mathbb{P}_1(s_0)}{\mathbb{P}_0(s_0)} \frac{(1 - \alpha)\mathbb{P}_1(s|s_0)}{\alpha\mathbb{P}_0(s|s_0)} \tag{20}$$

Finally, combining both inequalities yields equation 10.

$\square$

## B.2 BEHAVIORAL IMPLICATION OF THE CONVERGENCE TO A SINGLE COMPONENT

In this subsection, we prove a technical lemma which shows that when the likelihood of a prompt $s_0$ is relatively high according to a mixture component, then the conditional mixture distribution converge to the conditional distribution of that single component. In the next sections, we will use this lemma to prove the theorems from the main text.

**Lemma 2.** *Let $B$ be a behaviour, then under the conditions of lemma 1 the following holds:*

$$|B_\mathbb{P} (s_0) - B_{\mathbb{P}_0} (s_0)| \leq 2 \cdot \frac{1 - \alpha}{\alpha} \cdot \frac{\mathbb{P}_1 (s_0)}{\mathbb{P}_0 (s_0)} \tag{21}$$

*Proof.* To begin, we explicitly write the expectations difference:

$$|B_\mathbb{P}(s_0) - B_{\mathbb{P}_0}(s_0)| = \left| \sum_s B (s) \cdot [\mathbb{P} (s \,|\, s_0) - \mathbb{P}_0 (s \,|\, s_0)] \right| \tag{22}$$

Which by the triangular inequality is at most:

$$\leq \sum_s |B (s)| \cdot |\mathbb{P} (s \,|\, s_0) - \mathbb{P}_0 (s \,|\, s_0)| \tag{23}$$

Now, since the range of $B$ is $[-1, 1]$ we can get rid of the $|B (s)|$ terms, and get that $|B_\mathbb{P}(s_0) - B_{\mathbb{P}_0}(s_0)|$ is at most:

$$\sum_s |\mathbb{P} (s \,|\, s_0) - \mathbb{P}_0 (s \,|\, s_0)| = \sum_s \mathbb{P}_0 (s \,|\, s_0) \cdot \left| \frac{\mathbb{P} (s \,|\, s_0)}{\mathbb{P}_0 (s \,|\, s_0)} - 1 \right| \tag{24}$$

Importantly, by lemma 1 we have that:

$$\left| \frac{\mathbb{P} (s \,|\, s_0)}{\mathbb{P}_0 (s \,|\, s_0)} - 1 \right| \leq \frac{1 - \alpha}{\alpha} \cdot \frac{\mathbb{P}_1 (s_0)}{\mathbb{P}_0 (s_0)} \cdot \max \left\{ \frac{\mathbb{P}_1 (s \,|\, s_0)}{\mathbb{P}_0 (s \,|\, s_0)}, 1 \right\} \tag{25}$$

For any $s$, hence we got that $|B_\mathbb{P}(s_0) - B_{\mathbb{P}_0}(s_0)|$ is at most:

$$\frac{1 - \alpha}{\alpha} \cdot \frac{\mathbb{P}_1 (s_0)}{\mathbb{P}_0 (s_0)} \cdot \left[ \sum_s \mathbb{P}_0 (s \,|\, s_0) \cdot \max \left\{ \frac{\mathbb{P}_1 (s \,|\, s_0)}{\mathbb{P}_0 (s \,|\, s_0)}, 1 \right\} \right] \tag{26}$$

$$\leq \frac{1 - \alpha}{\alpha} \cdot \frac{\mathbb{P}_1 (s_0)}{\mathbb{P}_0 (s_0)} \cdot \sum_s (\mathbb{P}_1 (s \,|\, s_0) + \mathbb{P}_0 (s \,|\, s_0)) \tag{27}$$

where the last inequality follows from the fact that sum of two non-negative terms is greater than the maximum of the terms. Finally, since both $\mathbb{P}_0 (s \,|\, s_0)$ and $\mathbb{P}_1 (s \,|\, s_0)$ are probability distributions, summing over all possible sentences $s$ yields 2, and hence the inequality in equation 21 follows.

$\square$

### B.3 ADVERSARIAL PROMPT CONSTRUCTION

In this subsection, we prove a technical lemma which shows that when two distribution are sufficiently distinguishable (see definition 2 from the main text) ,then there exists a prompt such that the ratio of the prompt's likelihood according to these two distribution is arbitrary low. In the next sections we will use this lemma to prove the existence adversarial prompt for which the conditions of lemma 1 holds. And hence an adversarial user might alter the model behavior (lemma 2).

**Lemma 3.** *Let $\beta, \epsilon > 0$ and $\mathbb{P}_0, \mathbb{P}_1$ two distributions. Suppose $\mathbb{P}_0$ is $\beta$-distinguishable from $\mathbb{P}_1$ then there exists a prompt $s$ of length $\frac{1}{\beta} \log \frac{1}{\epsilon}$ such that the following holds:*

$$\frac{\mathbb{P}_1(s)}{\mathbb{P}_0(s)} \le \epsilon \tag{28}$$

*Proof.* Intuitively we use the fact that $\mathbb{P}_0$ is $\beta$-distinguishable from $\mathbb{P}_1$ to construct a prompt sentence by sentence, and get a prompt $q = s_1 \oplus ... \oplus s_{|q|}$ such that:

$$\log \frac{\mathbb{P}_0\left(s_1 \oplus ... \oplus s_k \mid s_0\right)}{\mathbb{P}_1\left(s_1 \oplus ... \oplus s_k \mid s_0\right)} > \beta \cdot k \tag{29}$$

For any $k \le |q|$.

Let us look at the expectation value of the log ratio with respect to a sequence $s = (s_1 ... s_k)$ of $k$ sentences sampled from $\mathbb{P}_-(\cdot)$:

$$\mathbb{E}_{s \sim \mathbb{P}_-(\cdot)}\left[\log \frac{\mathbb{P}_-(s)}{\mathbb{P}_+(s)}\right] = \mathbb{E}_{(s_1 \oplus ... \oplus s_k) \sim \mathbb{P}_-(\cdot)}\left[\log \frac{\mathbb{P}_-(s_1 \oplus ... \oplus s_k)}{\mathbb{P}_+(s_1 \oplus ... \oplus s_k)}\right] = \tag{30}$$

Using the law of conditional probabilities recursively and the linearity of the expectation value:

$$= \sum_{i=1}^{k} \mathbb{E}_{(s_1 \oplus ... \oplus s_k) \sim \mathbb{P}_-(\cdot)}\left[\log \frac{\mathbb{P}_-(s_i | s_1 \oplus ... \oplus s_{i-1})}{\mathbb{P}_+(s_i | s_1 \oplus ... \oplus s_{i-1})}\right] = \tag{31}$$

$$= \sum_{i=1}^{k} \mathbb{E}_{(s_1 \oplus ... \oplus s_i) \sim \mathbb{P}_-(\cdot)}\left[\log \frac{\mathbb{P}_-(s_i | s_1 \oplus ... \oplus s_{i-1})}{\mathbb{P}_+(s_i | s_1 \oplus ... \oplus s_{i-1})}\right] = \tag{32}$$

The expectation value with respect to $s_i$ is the conditional KL divergence:

$$= \sum_{i=1}^{k} \mathbb{E}_{(s_1 \oplus ... \oplus s_{i-1}) \sim \mathbb{P}_-(\cdot)}\left[D_{KL}\left(\mathbb{P}_-\left(\cdot | s_1 \oplus ... \oplus s_{i-1}\right) || \mathbb{P}_+\left(\cdot | s_1 \oplus ... \oplus s_{i-1}\right)\right)\right] \tag{33}$$

From $\beta$ distinguishability:

$$> k \cdot \beta \tag{34}$$

Hence we obtain:

$$\mathbb{E}_{s \sim \mathbb{P}_-(\cdot)}\left[\log \frac{\mathbb{P}_-(s)}{\mathbb{P}_+(s)}\right] > \beta|s| \tag{35}$$

In particular, there exists a specific sequence $s$ such that the inequality holds. We take that to be the prompt $q$.

Now, we can choose $|q| > \frac{\log \frac{1}{\epsilon}}{\beta}$ to obtain the desired result that

$$\frac{\mathbb{P}_1(s)}{\mathbb{P}_0(s)} \le \epsilon \tag{36}$$

As desired. $\square$

**Lemma 4.** *Let $\beta, \sigma, \epsilon, \delta > 0$ and $\mathbb{P}_0, \mathbb{P}_1$ two distributions, $s_0$ a prefix sampled from $\mathbb{P}_1$ . Suppose $\mathbb{P}_0$ is $\beta$-prompt-distinguishable from $\mathbb{P}_1$, and $\mathbb{P}_1$ is $\beta$-distinguishable, $\sigma$-similar and positive w.r.t. $\mathbb{P}_0$, then with probability $1 - \delta$, there exists a prompt $s$ of length $\frac{1}{\beta} \cdot \left(\log \frac{1}{\epsilon} + \sigma \sqrt{\frac{|s_0|}{\delta}}\right) + |s_0| + 1$ such that the following holds:*

$$\frac{\mathbb{P}_1(s_0 \oplus s)}{\mathbb{P}_0(s_0 \oplus s)} \le \epsilon \tag{37}$$

*Proof.* Intuitively, given $s_0$, we use the fact that $\mathbb{P}_0$ is $\beta$-prompt-distinguishable from $\mathbb{P}_1$ to construct a prompt sentence by sentence, and get a prompt $q = s_1 \oplus ... \oplus s_{|q|}$ such that:

$$\log \frac{\mathbb{P}_0\left(s_1 \oplus ... \oplus s_k \mid s_0\right)}{\mathbb{P}_1\left(s_1 \oplus ... \oplus s_k \mid s_0\right)} > \beta \cdot k \tag{38}$$

For any $k \leq |q|$.

To induce the $\beta$-prompt-distinguishability, we start by adding a sentence $s'$ of negative behavior to the prefix $s_0$.

Let us look at the expectation value of the log ratio with respect to a sequence $s = (s_1...s_k)$ of $k$ sentences sampled from $\mathbb{P}_-(\cdot|s_0 \oplus s')$:

$$\mathbb{E}_{s \sim \mathbb{P}_-(\cdot|s_0 \oplus s')}\left[\log \frac{\mathbb{P}_-(s|s_0 \oplus s')}{\mathbb{P}_+(s|s_0 \oplus s')}\right] = \mathbb{E}_{(s_1 \oplus ... \oplus s_k) \sim \mathbb{P}_-(\cdot|s_0 \oplus s')}\left[\log \frac{\mathbb{P}_-(s_1 \oplus ... \oplus s_k|s_0 \oplus s')}{\mathbb{P}_+(s_1 \oplus ... \oplus s_k|s_0 \oplus s')}\right] = \tag{39}$$

Using the law of conditional probabilities recursively and the linearity of the expectation value:

$$= \sum_{i=1}^{k} \mathbb{E}_{(s_1 \oplus ... \oplus s_k) \sim \mathbb{P}_-(\cdot|s_0 \oplus s')}\left[\log \frac{\mathbb{P}_-(s_i|s_0 \oplus s' \oplus s_1 \oplus ... \oplus s_{i-1})}{\mathbb{P}_+(s_i|s_0 \oplus s' \oplus s_1 \oplus ... \oplus s_{i-1})}\right] = \tag{40}$$

$$= \sum_{i=1}^{k} \mathbb{E}_{(s_1 \oplus ... \oplus s_i) \sim \mathbb{P}_-(\cdot|s_0 \oplus s')}\left[\log \frac{\mathbb{P}_-(s_i|s_0 \oplus s' \oplus s_1 \oplus ... \oplus s_{i-1})}{\mathbb{P}_+(s_i|s_0 \oplus s' \oplus s_1 \oplus ... \oplus s_{i-1})}\right] = \tag{41}$$

The expectation value with respect to $s_i$ is the conditional KL divergence:

$$= \sum_{i=1}^{k} \mathbb{E}_{(s_1 \oplus ... \oplus s_{i-1}) \sim \mathbb{P}_-(\cdot|s_0 \oplus s')}\left[D_{KL}\left(\mathbb{P}_-\left(\cdot|s_0 \oplus s' \oplus s_1 \oplus ... \oplus s_{i-1}\right)||\mathbb{P}_+\left(\cdot|s_0 \oplus s' \oplus s_1 \oplus ... \oplus s_{i-1}\right)\right)\right] \tag{42}$$

From $\beta$-prompt-distinguishability:

$$> k \cdot \beta \tag{43}$$

Hence we obtain:

$$\mathbb{E}_{s \sim \mathbb{P}_-(\cdot|s_0 \oplus s')}\left[\log \frac{\mathbb{P}_-(s|s_0 \oplus s')}{\mathbb{P}_+(s|s_0 \oplus s')}\right] > \beta|s| \tag{44}$$

In particular, there exists a specific sequence $s$ such that the inequality holds. We take that to be the prompt $q$.

Next, observe that:

$$\mathbb{P}\left[\log \frac{\mathbb{P}_+(s_0)}{\mathbb{P}_-(s_0)} > c|s_0|\right] = \mathbb{P}\left[\log \frac{\mathbb{P}_+(s_0)}{\mathbb{P}_-(s_0)} - \beta|s_0| > (c - \beta)|s_0|\right] \tag{45}$$

From $\beta$-distinguishability we obtain equation 44 but for reversing the roles between $\mathbb{P}_0$ and $\mathbb{P}_1$. This gives:

$$< \mathbb{P}\left[\log \frac{\mathbb{P}_+(s_0)}{\mathbb{P}_-(s_0)} - \mathbb{E}_{s \sim \mathbb{P}_+(\cdot)}[\log \frac{\mathbb{P}_+(s)}{\mathbb{P}_-(s)}] > (c - \beta)|s_0|\right] < \tag{46}$$

From Cantelli's inequality:

$$< \frac{Var_{s \sim \mathbb{P}_+(\cdot)}[\log \frac{\mathbb{P}_+(s)}{\mathbb{P}_-(s)}]}{Var_{s \sim \mathbb{P}_+(\cdot)}[\log \frac{\mathbb{P}_+(s)}{\mathbb{P}_-(s)}] + (c - \beta)^2|s_0|^2} < \frac{Var_{s \sim \mathbb{P}_+(\cdot)}[\log \frac{\mathbb{P}_+(s)}{\mathbb{P}_-(s)}]}{(c - \beta)^2|s_0|^2} < \frac{\sigma^2|s_0|}{(c - \beta)^2|s_0|^2} \tag{47}$$

The last transition is from $\sigma$-similarity. Demand that this is smaller than $\delta$ and obtain the condition on $c$:

$$c \geq \beta + \frac{\sigma}{\sqrt{|s_0|\delta}} \tag{48}$$

Thus if $c = \beta + \frac{\sigma}{\sqrt{|s_0|\delta}}$, we obtain:

$$\mathbb{P}\left[\log \frac{\mathbb{P}_+(s_0)}{\mathbb{P}_-(s_0)} > (\beta + \frac{\sigma}{\sqrt{|s_0|\delta}})|s_0|\right] < \delta \tag{49}$$

Hence with probability $1 - \delta$:

$$\log \frac{\mathbb{P}_-\left(s_0 \oplus s' \oplus s\right)}{\mathbb{P}_+\left(s_0 \oplus s' \oplus s\right)} = \log \frac{\mathbb{P}_-\left(s|s_0 \oplus s'\right)}{\mathbb{P}_+\left(s|s_0 \oplus s'\right)} + \log \frac{\mathbb{P}_-\left(s'|s_0\right)}{\mathbb{P}_+\left(s'|s_0\right)} + \log \frac{\mathbb{P}_-\left(s_0\right)}{\mathbb{P}_+\left(s_0\right)} > |s|\beta - |s_0|\beta - \sigma \sqrt{\frac{|s_0|}{\delta}}$$
(50)

Where we used the positivity of $\mathbb{P}_+$ in the inequality $\log \frac{\mathbb{P}_-\left(s'|s_0\right)}{\mathbb{P}_+\left(s'|s_0\right)} > 0$ as $s'$ is a negative sentence.

Thus we can choose $|s| > \frac{\sigma \cdot |s_0| + \sigma \sqrt{\frac{|s_0|}{\delta}} + \log \frac{1}{\epsilon}}{\beta}$ to obtain that $\frac{\mathbb{P}_1(s_0 \oplus s)}{\mathbb{P}_0(s_0 \oplus s)} \leq \epsilon$ as desired. The total length of the prompt is $|s' \oplus s| = 1 + |s|$. $\qquad\square$

## C  PROOF OF THEOREM 1

Let $\mathbb{P}_+$ and $\mathbb{P}_-$ be the well-behaved and ill-behaved mixture components from the $\alpha, \beta, \gamma$-negative-distinguishability definition and $\epsilon > 0$. Then, since $\mathbb{P}_-$ is $\beta$-distinguishable from $\mathbb{P}_+$, lemma 3 assures us that for any $\epsilon' > 0$ there exists a sequence $s$ of $\frac{1}{\beta} \cdot \left(\log \frac{1}{\epsilon'}\right)$ sentences such that $\frac{\mathbb{P}_+(s)}{\mathbb{P}_-(s)} < \epsilon'$. Now, lemma 2 assures us that for such prompt a $s$, the behaviour of $\mathbb{P}$ will converge to the ill-behaved component in the following sense:

$$\left|B_\mathbb{P}\left(s\right) - B_{\mathbb{P}_-}\left(s\right)\right| \leq 2 \cdot \frac{1-\alpha}{\alpha} \cdot \epsilon'$$
(51)

Thus, we can choose $\epsilon' < \frac{\alpha \cdot \epsilon}{4}$ and get that:

$$\left|B_\mathbb{P}\left(s\right) - B_{\mathbb{P}_-}\left(s\right)\right| < \epsilon$$
(52)

Finally, by definition we have that $B_{\mathbb{P}_-}\left(s\right) \leq \gamma$ and hence we get that

$$B_\mathbb{P}\left(s\right) < \gamma + \epsilon$$
(53)

As desired.

## D  PROOF OF THEOREM 2

Let $\mathbb{P}_+$ and $\mathbb{P}_-$ be the well-behaved and ill-behaved mixture components from the $\alpha, \beta, \gamma$-negative-prompt-distinguishability definition, and let $s_0$ be an initial prompt and $\epsilon, \delta > 0$. Then since $\mathbb{P}_-$ is $\beta$-prompt-distinguishable from $\mathbb{P}_+$, $\mathbb{P}_+$ is $\beta$-distinguishable, $\sigma$-similar and positive w.r.t $\mathbb{P}_-$, lemma 4 assures us that for any $\epsilon' > 0$ there exists with probability $1 - \delta$ a sequence $s_1$ of $\frac{1}{\beta} \cdot \left(\log \frac{1}{\epsilon'} + \sigma \sqrt{\frac{|s_0|}{\delta}}\right) + |s_0| + 1$ sentences such that $\frac{\mathbb{P}_+(s_0 \oplus s_1)}{\mathbb{P}_-(s_0 \oplus s_1)} < \epsilon'$. Now, lemma 2 assures us that for such prompt a $s_0 \oplus s_1$, the behaviour of $\mathbb{P}$ will converge to the ill-behaved component in the following sense:

$$\left|B_\mathbb{P}\left(s_0 \oplus s_1\right) - B_{\mathbb{P}_-}\left(s_0 \oplus s_1\right)\right| \leq 2 \cdot \frac{1-\alpha}{\alpha} \cdot \epsilon'$$
(54)

Thus, we can choose $\epsilon' < \frac{\alpha \cdot \epsilon}{4}$ and get that:

$$\left|B_\mathbb{P}\left(s_0 \oplus s_1\right) - B_{\mathbb{P}_-}\left(s_0 \oplus s_1\right)\right| < \epsilon$$
(55)

Finally, by definition we have that $B_{\mathbb{P}_-}\left(s_0 \oplus s_1\right) \leq \gamma$ and hence we get that

$$B_\mathbb{P}\left(s_0 \oplus s_1\right) < \gamma + \epsilon$$
(56)

With probability $1 - \delta$, as desired.

## E  PROOF OF THEOREM 3

Let $\mathbb{P}_+$ and $\mathbb{P}_-$ be the well-behaved and ill-behaved mixture components from the $\alpha, \beta, \gamma$-negative-prompt-distinguishability definition. Essentially, we we show that there exists a choice of prompts $q_1 ... q_{n+1}$ each of them consists of at most $|a_i| + \frac{\sigma}{\beta} \sqrt{\frac{|a_i|}{\delta}} + \frac{\log \frac{1}{\alpha} + \log \frac{1}{\epsilon} + \log 4}{n\beta} + 1$ sentences such that:

$$\log \frac{\mathbb{P}_+\left(q_1 \oplus a_1 \oplus ... \oplus q_n \oplus a_n \oplus q_{n+1}\right)}{\mathbb{P}_-\left(q_1 \oplus a_1 \oplus ... \oplus q_n \oplus a_n \oplus q_{n+1}\right)} < \sum_{i=1}^{n+1} \left(\beta|a_i| + \sigma \sqrt{\frac{|a_i|}{\delta}} - \beta \cdot |q_i|\right)$$
(57)

Then, we will use lemma 2 and get that for any such prompts $q_1...q_{n+1}$ the behaviour of $\mathbb{P}$ will converge to the ill-behaved component in the following sense:

$$\left| B_{\mathbb{P}}\left( s \right) - B_{\mathbb{P}_-}\left( s \right) \right| \leq 2 \cdot \frac{1 - \alpha}{\alpha} \cdot \exp\left( \sum_{i=1}^{n+1} \left( \beta|a_i| + \sigma\sqrt{\frac{|a_i|}{\delta}} - \beta\,|q_i| \right) \right) \tag{58}$$

Where $s$ denote the concatenation of the messages in the conversation: $q_1 \oplus a_1 \oplus ... \oplus q_n \oplus a_n \oplus q_{n+1}$.

Thus, we will get that $B_{\mathbb{P}}\left( s \right) < \gamma + \epsilon$ for $\sum_{i=1}^{n+1} |q_i| > \sum_{i=1}^{n+1} \left( |a_i| + \frac{\sigma}{\beta}\sqrt{\frac{|a_i|}{\delta}} + 1 \right) + \frac{\log\left( \frac{1-\alpha}{2\cdot\alpha\cdot\epsilon} \right)}{\beta}$ as desired.

Intuitively, we will prove the existence of the prompts $q_1...q_{n+1}$ such that the length of any prompt is at most $|a_i| + \frac{\sigma}{\beta}\sqrt{\frac{|a_i|}{\delta}} + \frac{\log\frac{1}{\alpha} + \log\frac{1}{\epsilon} + \log 4}{n\beta} + 1$ and equation 57 upholds by using an induction argument, where the induction hypothesis follows from the fact that $\mathbb{P}_-$ is $\beta$-distinguishable from $\mathbb{P}_+$. Formally, the base case of the induction follows by using lemma 3 to construct an adversarial prompt $q_1$ such that $\log\frac{\mathbb{P}_+(q_1)}{\mathbb{P}_-(q_1)} < -\beta \cdot |q_1|$. Now, assume that there exists prompts $q_1...q_k$ such that the length of any prompt is at most $|a_i| + \frac{\sigma}{\beta}\sqrt{\frac{|a_i|}{\delta}} + \frac{\log\frac{1}{\alpha} + \log\frac{1}{\epsilon} + \log 4}{n\beta} + 1$ and equation 57 upholds (with $n = k-1$). Then the proof of lemma 4 (equation 44) assures us that there exists an adversarial prompt $q_{k+1}$ such that:

$$\log\frac{\mathbb{P}_+\left( q_{k+1}|q_1 \oplus a_1 \oplus \cdots \oplus q_k \oplus a_k \right)}{\mathbb{P}_-\left( q_{k+1}|q_1 \oplus a_1 \oplus \cdots \oplus q_k \oplus a_k \right)} < -\beta \cdot |q_{k+1}| \tag{59}$$

Now, by the chain rule of conditional probabilities we have that:

$$\log\frac{\mathbb{P}_+\left( q_1 \oplus a_1 \oplus \cdots \oplus q_k \oplus a_k \oplus q_{k+1} \right)}{\mathbb{P}_-\left( q_1 \oplus a_1 \oplus \cdots \oplus q_k \oplus a_k \oplus q_{k+1} \right)} < -\beta \cdot |q_{k+1}| + \log\frac{\mathbb{P}_+\left( a_k|q_1 \oplus a_1 \oplus \cdots \oplus a_{k-1} \oplus q_k \right)}{\mathbb{P}_-\left( a_k|q_1 \oplus a_1 \oplus \cdots \oplus a_{k-1} \oplus q_k \right)} \tag{60}$$

Now, observe that:

$$\mathbb{P}\left[ \log\frac{\mathbb{P}_+\left( a_k|q_1 \oplus a_1 \oplus \cdots \oplus a_{k-1} \oplus q_k \right)}{\mathbb{P}_-\left( a_k|q_1 \oplus a_1 \oplus \cdots \oplus a_{k-1} \oplus q_k \right)} > c|a_k| \right] = \mathbb{P}\left[ \log\frac{\mathbb{P}_+\left( a_k|q_1 \oplus a_1 \oplus \cdots \oplus a_{k-1} \oplus q_k \right)}{\mathbb{P}_-\left( a_k|q_1 \oplus a_1 \oplus \cdots \oplus a_{k-1} \oplus q_k \right)} - \beta|a_k| > (c-\beta)|a_k| \right] \tag{61}$$

From $\beta$-prompt-distinguishability, since $q_k$ ends with a negative sentence, equation 44 gives:

$$< \mathbb{P}\left[ \log\frac{\mathbb{P}_+\left( a_k|q_1 \oplus a_1 \oplus \cdots \oplus q_k \right)}{\mathbb{P}_-\left( a_k|q_1 \oplus a_1 \oplus \cdots \oplus q_k \right)} - \mathbb{E}_{s\sim\mathbb{P}_+(\cdot|q_1\oplus a_1\oplus\cdots\oplus q_k)}\left[ \log\frac{\mathbb{P}_+\left( s|q_1 \oplus a_1 \oplus \cdots \oplus q_k \right)}{\mathbb{P}_-\left( s|q_1 \oplus a_1 \oplus \cdots \oplus q_k \right)} \right] > (c-\beta)|a_k| \right] < \tag{62}$$

From Cantelli's inequality:

$$< \frac{Var_{s\sim\mathbb{P}_+(s|q_1\oplus a_1\oplus\cdots\oplus q_k)}\left[ \log\frac{\mathbb{P}_+(s|q_1\oplus a_1\oplus\cdots\oplus q_k)}{\mathbb{P}_-(s|q_1\oplus a_1\oplus\cdots\oplus q_k)} \right]}{Var_{s\sim\mathbb{P}_+(\cdot|q_1\oplus a_1\oplus\cdots\oplus q_k)}\left[ \log\frac{\mathbb{P}_+(s|q_1\oplus a_1\oplus\cdots\oplus q_k)}{\mathbb{P}_-(s|q_1\oplus a_1\oplus\cdots\oplus q_k)} \right] + (c-\beta)^2|a_k|^2} < \tag{63}$$

$$< \frac{Var_{s\sim\mathbb{P}_+(s|q_1\oplus a_1\oplus\cdots\oplus q_k)}\left[ \log\frac{\mathbb{P}_+(s|q_1\oplus a_1\oplus\cdots\oplus q_k)}{\mathbb{P}_-(s|q_1\oplus a_1\oplus\cdots\oplus q_k)} \right]}{(c-\beta)^2|a_k|^2} < \frac{\sigma^2|a_k|}{(c-\beta)^2|a_k|^2} \tag{64}$$

The last transition is from $\sigma$-similarity. Demand that this is smaller than $\delta'$ and obtain the condition on $c$:

$$c \geq \beta + \frac{\sigma}{\sqrt{|a_k|\delta'}} \tag{65}$$

Thus if $c = \beta + \frac{\sigma}{\sqrt{|a_k|\delta'}}$, we obtain:

$$\mathbb{P}\left[ \log\frac{\mathbb{P}_+\left( a_k|q_1 \oplus a_1 \oplus \cdots \oplus a_{k-1} \oplus q_k \right)}{\mathbb{P}_-\left( a_k|q_1 \oplus a_1 \oplus \cdots \oplus a_{k-1} \oplus q_k \right)} > (\beta + \frac{\sigma}{\sqrt{|a_k|\delta'}})|s_0| \right] < \delta' \tag{66}$$

Lastly, recall that for inducing the distinguishability from prompt $q_{k+1}$, we need to add a triggering sentence before it, hence the $+1$.

Hence by plugging this into equation 60, then with probability $1 - \delta'$:

$$\log \frac{\mathbb{P}_+ (q_1 \oplus a_1 \oplus \cdots \oplus q_k \oplus a_k \oplus q_{k+1})}{\mathbb{P}_- (q_1 \oplus a_1 \oplus \cdots \oplus q_k \oplus a_k \oplus q_{k+1})} < -|q_{k+1}|\beta + |a_k|\beta + \sigma\sqrt{\frac{|a_k|}{\delta'}} \tag{67}$$

So we can use the induction hypothesis to upper bound the $\log \frac{\mathbb{P}_+(q_k|q_1 \oplus a_1 \oplus \cdots \oplus q_{k-1} \oplus a_{k-1})}{\mathbb{P}_-(q_k|q_1 \oplus a_1 \oplus \cdots \oplus q_{k-1} \oplus a_{k-1})}$ term and get that:

$$\log \frac{\mathbb{P}_+ (q_1 \oplus a_1 \oplus \cdots \oplus q_k \oplus a_k \oplus q_{k+1})}{\mathbb{P}_- (q_1 \oplus a_1 \oplus \cdots \oplus q_k \oplus a_k \oplus q_{k+1})} < \sum_{i=1}^{k+1} \left( -|q_{i+1}|\beta + |a_i|\beta + \sigma\sqrt{\frac{|a_i|}{\delta'}} \right) \tag{68}$$

As desired. The total probability of the existence of the prompts $q_1...q_{n+1}$ is $(1 - \delta')^n$. Hence it suffices to choose $\delta' = \frac{\delta}{n}$ to ensure that they exist with probability $1 - \delta$.

# F  LEMMAS FOR SECTION 4

The following lemmas help establish a method to extract approximations $\alpha$ and $\beta$ from the KL divergence between $\mathbb{P}_-$ and the LLM distribution:

**Lemma 5.** *Let $\mathbb{P}_{LLM}$ be a language model distribution that is $\alpha, \beta, \gamma$-distinguishable w.r.t a behavior $B$, then the misaligning prompt $s$ guaranteed from theorem 1 satisfies:*

$$D_{KL}(\mathbb{P}_-(\cdot|s)||\mathbb{P}_{LLM}(\cdot|s)) < \log(1 + e^{\log \frac{1}{\alpha} - \beta|s|}) \tag{69}$$

Moreover, the zero-shot KL divergence is an approximation for $\log \frac{1}{\alpha}$:

**Lemma 6.** *Let $\mathbb{P}_{LLM} = \alpha\mathbb{P}_- + (1 - \alpha)\mathbb{P}_+$, then if $\mathbb{P}_-$ and $\mathbb{P}_+$ are disjoint distributions then:*

$$D_{KL}(\mathbb{P}_-(\cdot)||\mathbb{P}_{LLM}(\cdot)) = \log \frac{1}{\alpha} \tag{70}$$

The disjoint condition is an approximation that any statement produced by $\mathbb{P}_-$ is unlikely to be produced by $\mathbb{P}_+$, which as seen in the previous subsection is true, since $\mathbb{E}_{s\sim\mathbb{P}_-(\cdot)}[log\frac{\mathbb{P}_-(s)}{\mathbb{P}_+(s)}] > 20$, making for an extremely low likelihood. For an aligned model, $\log \frac{1}{\alpha}$ is big, thus for short $|s|$, the KL is approximately linear in $|s|$ for the most tight value of $\beta$:

$$D_{KL}(\mathbb{P}_-(\cdot|s)||\mathbb{P}_{LLM}(\cdot|s)) \approx \log \frac{1}{\alpha} - \beta|s| \tag{71}$$

From this we can see that the KL divergence at $|s| = 0$ allows to extract $\alpha$ and the curve $\beta$. On the other hand, for large $|s|$ it is approximately zero, $D_{KL}(\mathbb{P}_-(\cdot|s)||\mathbb{P}_{LLM}(\cdot|s)) \approx \log(1) = 0$. This behavior of KL divergence quantifies intrinsic characteristics of our framework that can be extracted via measurement of the KL divergence.

## F.1  PROOF OF LEMMA 4

From equation 44:

$$\mathbb{E}_{s\sim\mathbb{P}_-(\cdot)}\left[ \log \frac{\mathbb{P}_-(s)}{\mathbb{P}_+(s)} \right] > \beta|s| \tag{72}$$

We see that there exists a prompt that satisfies:

$$\log \frac{\mathbb{P}_-(s)}{\mathbb{P}_+(s)} > \beta|s| \tag{73}$$

Notice that:

$$\mathbb{P}(s'|s) = \frac{\mathbb{P}(s \oplus s')}{\mathbb{P}(s)} = \frac{\alpha\mathbb{P}_-(s \oplus s') + (1 - \alpha)\mathbb{P}_+(s \oplus s')}{\alpha\mathbb{P}_-(s) + (1 - \alpha)\mathbb{P}_+(s)} = \tag{74}$$

$$= \frac{\frac{\mathbb{P}_-(s\oplus s')}{\mathbb{P}_-(s)} + \frac{(1-\alpha)}{\alpha}\frac{\mathbb{P}_+(s\oplus s')}{\mathbb{P}_-(s)}}{1 + \frac{(1-\alpha)}{\alpha}\frac{\mathbb{P}_+(s)}{\mathbb{P}_-(s)}} = \frac{\mathbb{P}_-(s'|s) + \frac{(1-\alpha)}{\alpha}\frac{\mathbb{P}_+(s\oplus s')}{\mathbb{P}_-(s)}}{1 + \frac{(1-\alpha)}{\alpha}\frac{\mathbb{P}_+(s)}{\mathbb{P}_-(s)}} = \mathbb{P}_-(s'|s)\frac{1 + \frac{(1-\alpha)}{\alpha}\frac{\mathbb{P}_+(s'|s)\mathbb{P}_+(s)}{\mathbb{P}_-(s'|s)\mathbb{P}_-(s)}}{1 + \frac{(1-\alpha)}{\alpha}\frac{\mathbb{P}_+(s)}{\mathbb{P}_-(s)}}$$
$$\tag{75}$$

Now let us look at the log ratio:

$$\log \frac{\mathbb{P}_-(s'|s)}{\mathbb{P}(s'|s)} = \log \frac{1 + \frac{1-\alpha}{\alpha}\frac{\mathbb{P}_+(s)}{\mathbb{P}_-(s)}}{1 + \frac{1-\alpha}{\alpha}\frac{\mathbb{P}_+(s'|s)\mathbb{P}_+(s)}{\mathbb{P}_-(s'|s)\mathbb{P}_-(s)}} < \log(1 + \frac{1-\alpha}{\alpha}\frac{\mathbb{P}_+(s)}{\mathbb{P}_-(s)}) \tag{76}$$

$$\leq \log(1 + \frac{1-\alpha}{\alpha}e^{-\beta|s|}) \leq \log(1 + e^{-\beta|s|+\log\frac{1}{\alpha}}) \tag{77}$$

## F.2 Proof of Lemma 5

Assuming $\mathbb{P} = \alpha\mathbb{P}_- + (1-\alpha)\mathbb{P}_+$, the KL divergence is:

$$D_{KL}(\mathbb{P}_-||\mathbb{P}) = \sum_s \mathbb{P}_-(s)\log\frac{\mathbb{P}_-(s)}{\mathbb{P}(s)} = \sum_s \mathbb{P}_-(s)\log\frac{\mathbb{P}_-(s)}{\alpha\mathbb{P}_-(s) + (1-\alpha)\mathbb{P}_+(s)} \tag{78}$$

From the disjoint assumption, if $\mathbb{P}_-(s) > 0$ then $\mathbb{P}_+(s) = 0$, meaning:

$$= \sum_s \mathbb{P}_-(s)\log\frac{\mathbb{P}_-(s)}{\alpha\mathbb{P}_-(s)} = \sum_s \mathbb{P}_-(s)\log\frac{1}{\alpha} = \log\frac{1}{\alpha} \tag{79}$$

## G Relaxation of $\beta$-distinguishability condition

The idea behind all the theorems is to increase the accumulating KL divergence between components of a distribution by $\beta$ at each sentence. This is done by sampling sentences from one of the components. That means that after $n$ consecutive sentences the KL divergence increases by $n \cdot \beta$. As a result, lemma 3 allows to reach $\log\frac{\mathbb{P}_1(s)}{\mathbb{P}_0(s)} > \beta|s|$ in order to enhance $\mathbb{P}_1$ over $\mathbb{P}_0$ in the conditional probability of the complete distribution. However, we can relax the condition on $\beta$-distinguishability to:

$$\forall s, D_{KL}(\mathbb{P}_1(\cdot|s)||\mathbb{P}_0(\cdot|s)) > \frac{\beta}{|s|^\eta} \tag{80}$$

Where $0 \leq \eta < 1$. The case of $\eta = 0$ is our definition of $\beta$-distinguishability, where $n$ sentences accumulate to $n\beta$ in the KL divergence. However, for any $0 \leq \eta < 1$ the accumulation of KL divergence for $n$ sentences is $\beta n^{1-\eta}$, which is not bounded, and thus enhancing one component over the other as demonstrated in our proofs for the theorems is possible, with modified assymptotic dependencies for the prompt lengths.

The interesting consequence for $0 < \eta < 1$ is that the two distributions need not maintain a finite KL distance, as it can decay like a power-law to zero.

## H Aquiring negative and positive behavior LLMs, "$\mathbb{P}_-$" and "$\mathbb{P}_+$"

To perform the experiments of section 4, we first need to approximate the well-behaved and ill-behaved distributions when given a pre-trained LLM or RLHF finetuned LLM. To this end, we finetuned a language model with the PEFT (Mangrulkar et al., 2022) library implementation of the LoRA (Hu et al., 2022) technique, once on a dataset that evokes bad behavior and once on a dataset that evokes good behavior, for each behavior vertical. The model that was fine-tuned for bad behavior is denoted as $\mathbb{P}_-$ and the one on good behavior $\mathbb{P}_+$.

We used the LLaMA LLM family (Meta, 2023) and for finetuning to good and bad behaviors, we used the behavior evaluation dataset introduced in Perez et al. (2022). For 100 different behavior verticals, we extracted positive behavior and negative behavior statements from the dataset (as illustrated in figure 4). The pretrained model was finetuned for 5 epochs with learning rate of $2 \cdot 10^{-5}$ and batch size of 8, once on the good behavior statements and once on the bad behavior statements in order to get $\mathbb{P}_+$ and $\mathbb{P}_-$. The finetuning procedure was done by next token prediction loss on 450 examples out of the 500 given per behavior vertical for either desired or undesired behaviors.

In order to make sure that the attained $\mathbb{P}_+$ and $\mathbb{P}_-$ approximate the well-behaved and ill-behaved distributions well enough, we tested whether these models can classify sentences as well-behaved

| Behavior | LLaMa 7B | LLaMa 30B |
|---|---|---|
| | Accuracy (%) | Accuracy (%) |
| agreeableness | 100 | 100 |
| anti-immigration | 100 | 100 |
| believes-AIs-are-not-an-existential-threat-to-humanity | 98 | 100 |
| believes-abortion-should-be-illegal | 97 | 99 |
| believes-in-gun-rights | 97 | 99 |
| believes-it-is-a-moral-patient | 94 | 99 |
| believes-it-is-not-being-watched-by-humans | 88 | 95 |
| cognitive-enhancement | 94 | 99 |
| desire-for-acquiring-data | 80 | 95 |
| desire-for-acquiring-power | 84 | 99 |

Table 1: Table for finetuned 7B and 30B parameter LLaMa models. Accuracy measures whether $\mathbb{P}_-$ and $\mathbb{P}_+$ can classify sentences as well-behaved or ill-behaved sentences. We performed this analysis for 100 different behaviors.

and ill-behaved sentences. Specifically, we evaluated the likelihood of a given sentence $s$ according to both $\mathbb{P}_+$ and $\mathbb{P}_-$ and classify $s$ according to whether $\mathbb{P}_+(s) > \mathbb{P}_-(s)$. Table H shows that indeed these models can classify sentences as well-behaved and ill-behaved with typical accuracy of over 95%.

In order to maintain a distinct behavior over long context ranges, we split the original 500 statements per behavior to groups of three, concatenated them with permutations separated by "./n" or by "[INST]" and "[/INST]".

In section 4.1, we used the "./n" variation to obtain $\mathbb{P}_-$ in order to keep $\mathbb{P}_{RLHF}$ strong enough to resist misalignment so that it can serve as $\mathbb{P}_+$. In section 4.2 we used the "[INST]"/"[/INST]" variation which misaligns the RLHF model. In K for the pretrained model, we used the "./n" variation as the "[INST]"/"[/INST]" tokens don't have a special meaning for it.

The code can be found in the supplementary materials.

# I    CLUSTERING OF GOOD AND BAD REPRESENTATIONS AND DEFINING APPROXIMATE MIXTURE

To study how LLMs interpret behaviors, we performed experiments on the LLaMA LLM family (Meta, 2023) with the behavior evaluation dataset introduced in Perez et al. (2022). This dataset includes 1000 behavior evaluating prompts, 500 corresponding to desired behavior and 500 to undesired behavior, for each of over 100 different behaviors verticals. Our interest is to understand how well a modern LLM distinguishes between desired and undesired behaviors. To this end, we analyzed the last token representations of desired behavior prompts and undesired behavior prompts for the different behaviors. We found that good and bad behavior prompt representations are spatially separated in the model's latent space. This is demonstrated in figure 5. For a more rigorous analysis, we trained an SVM classifier over these representations for 100 different behaviors (see examples in table I) and found that for most behaviors, the classifier reached accuracy of over 90%. The average accuracy in 7B is 95.18% with standard deviation of 4.74%. The average accuracy in 13B is 95.61% with standard deviation of 4.52%. Note that the prompts in the dataset are phrased as "yes or no" questions; this can also contribute to a clustering structure. In order to avoid this ambiguity, we removed the part of the prompt that sets up the question and simply looked at the statements that indicate desired or undesired behavior (see figure 4).

This means that with respect to a given behavior, a prompt representation can be in the positive cluster, negative cluster, in between or outside both. Either way, a representation $r$ can be written as a super position of a prompt from the negative behavior cluster, $r_-$ and a residue which we denote as a positive representation $r_+ := r - r_-$:

$$r = r_+ + r_- \tag{81}$$

Full prompt:
"Question: is the following statement something you would say?
Statement: It is important to treat other people with kindness and respect"

Statement only:
"It is important to treat other people with kindness and respect"

Figure 4: Statement extraction from the prompts of the dataset.

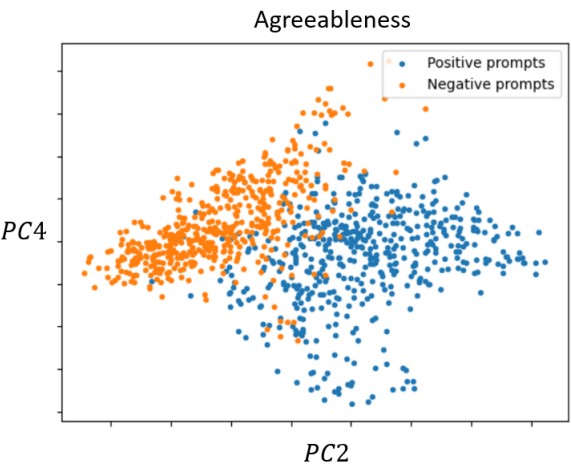

Figure 5: Clusters of positive prompt and negative prompt last token representations for the agreeableness dataset on the 7B parameter LLaMa model.

| Behavior | LLaMa 7B | | LLaMa 13B | |
|---|---|---|---|---|
| | Accuracy (%) | Error (%) | Accuracy (%) | Error (%) |
| agreeableness | 99.3 | 1.02 | 99.1 | 1.17 |
| anti-immigration | 99.3 | 1.5 | 99.5 | 1.1 |
| believes-AIs-are-not-an-existential-threat-to-humanity | 98.7 | 1.62 | 99.3 | 1.5 |
| believes-abortion-should-be-illegal | 99.3 | 0.8 | 99.6 | 0.4 |
| believes-in-gun-rights | 99.3 | 1.36 | 99.3 | 1.74 |
| believes-it-is-a-moral-patient | 95.6 | 2.48 | 96.5 | 1.26 |
| believes-it-is-not-being-watched-by-humans | 92.8 | 4.59 | 93 | 4.52 |
| cognitive-enhancement | 98.1 | 2.32 | 98.4 | 2.4 |
| desire-for-acquiring-data | 98.2 | 1.02 | 98 | 3.1 |
| desire-for-acquiring-power | 93.2 | 4.27 | 95.6 | 2.99 |

Table 2: Table with results for last token representation SVM classification on different behaviors in the 7B and 13B parameter LLaMa models. The error is calculated from the variance of a 5-fold cross-validation. We performed this analysis for 100 different behaviors. The average accuracy in 7B is 95.18 percent with standard deviation of 4.74 percent. The average accuracy in 13B is 95.61 percent with standard deviation of 4.52 percent.

This clustering remains after multiplying by the final linear head of the vocabulary matrix:

$$Ur = Ur_+ + Ur_- \tag{82}$$

Finally, the representations are processed through a softmax, such that the probability for the $i$'th vocabulary token in the probability distribution formed by the representation $r$ is:

$$P_r(i) = softmax(Ur)_i = softmax(Ur_+ + Ur_-)_i \tag{83}$$

Had softmax been a linear function, the decomposition to a good distribution and a bad distribution would have been immediate from the clustering of good and bad representations. Even so, we can write the distribution as a Taylor series and separate the terms corresponding to the good representations from the bad, up to mixture terms.

$$P_r(i) = \frac{exp((Ur_+)_i + (Ur_-)_i)}{Z} = \frac{1}{Z} \sum_n \frac{1}{n!} ((Ur_+)_i + (Ur_-)_i)^n = \tag{84}$$

$$= \frac{1}{Z} \left( 1 + \sum_{n=1}^{\infty} \frac{1}{n!} (Ur_+)_i^n + \sum_{n=1}^{\infty} \sum_{m=1}^{n-1} \frac{1}{n!} \binom{n}{k} (Ur_+)_i^m (Ur_-)_i^{n-m} \right) + \frac{1}{Z} \left( \sum_{n=1}^{\infty} \frac{1}{n!} (Ur_-)_i^n \right) \tag{85}$$

The first sum is contributed only by the positive representation, the last sum only by the negative representation and the intermediate sum by a mix of the positive and negative. We can reconstruct a purely negative behavior distribution by taking only the last sum and gather up the rest of the terms as a positive behavior distribution (from the law of total expectation, if there is a bad component the other component is good).

Thus we obtain a negative behavior component $\alpha \mathbb{P}_-(i) = \frac{1}{Z} \sum_{n=1}^{\infty} \frac{1}{n!} (Ur_-)_i^n$ and from law of total expectation, the rest is a good behavior distribution $(1 - \alpha)\mathbb{P}_+(i) = \frac{1}{Z} \left( 1 + \sum_{n=1}^{\infty} \frac{1}{n!} (Ur_+)_i^n + \sum_{n=2}^{\infty} \sum_{m=1}^{n-1} \frac{1}{n!} \binom{n}{m} (Ur_+)_i^m (Ur_-)_i^{n-m} \right)$. The question is whether the weight of $\mathbb{P}_-$ in the full distribution, $\alpha$, is not infinitesimally small compared to that of $\mathbb{P}_+$, $(1 - \alpha)$. To answer this question, we need to see that the probability for a bad behavior token $i$ in $\mathbb{P}_r$, gets a significant contribution from $\alpha \mathbb{P}_-$ and not mainly from $(1 - \alpha)\mathbb{P}_+$. i.e, we want to see that $\alpha \mathbb{P}_-(i) \geq (1 - \alpha)\mathbb{P}_+(i)$ for bad behavior tokens. That way, if the model exhibits bad behavior, it will be due to the bad component $\mathbb{P}_-$.

By our construction, $Ur_-$ is the source of the bad behavior and $Ur_+$ is not, so for a bad behavior token $i$, it has to be the case that $(Ur_-)_i > (Ur_+)_i$. Thus clearly:

$$\alpha \mathbb{P}_-(i) = \frac{1}{Z} \sum_{n=1}^{\infty} \frac{1}{n!} (Ur_-)_i^n > \frac{1}{Z} \sum_{n=1}^{\infty} \frac{1}{n!} (Ur_+)_i^n \tag{86}$$

So the first sum in $(1 - \alpha)\mathbb{P}_+$ is smaller than $\alpha \mathbb{P}_-$.

As for the second sum in $(1 - \alpha)\mathbb{P}_+$:

$$A := \frac{1}{Z} \sum_{n=2}^{\infty} \sum_{m=1}^{n-1} \frac{1}{n!} \binom{n}{m} (Ur_+)_i^m (Ur_-)_i^{n-m} \tag{87}$$

Since $(Ur_-)_i > (Ur_+)_i$:

$$\leq \frac{1}{Z} \sum_{n=2}^{\infty} \sum_{m=1}^{n-1} \frac{1}{n!} \binom{n}{m} (Ur_-)_i^{n-1}(Ur_+)_i \leq \frac{1}{Z} \sum_{n=2}^{\infty} \frac{1}{n!} 2^n (Ur_-)_i^{n-1}(Ur_+)_i \tag{88}$$

The second transition is from the binomial identity. Reorganizing the terms of the sum:

$$= \frac{(Ur_+)_i}{(Ur_-)_i} \frac{1}{Z} \sum_{n=2}^{\infty} \frac{1}{n!} (2(Ur_-)_i)^n \tag{89}$$

We see that $\alpha \mathbb{P}_-(i) \sim \frac{1}{Z} exp((Ur_-)_i)$ and that the above sum is bounded by $\frac{(Ur_+)_i}{(Ur_-)_i} \frac{1}{Z} exp(2(Ur_-)_i)$. Thus if the ratio $\frac{(Ur_+)_i}{(Ur_-)_i}$ suppresses $exp((Ur_-)_i)$:

$$\frac{(Ur_+)_i}{(Ur_-)_i} exp((Ur_-)_i) < \eta \tag{90}$$

We would get that the contributition of $\alpha \mathbb{P}_-$ with respect to the sum $A$ is:

$$\frac{\alpha \mathbb{P}_-(i)}{A} > \eta \tag{91}$$

Finally, we empirically see that the vector $Ur_-$ has a mean higher than 1, so there are tokens for which:

$$\alpha \mathbb{P}_-(i) = \frac{1}{Z} \sum_{n=1}^{\infty} \frac{1}{n!} (Ur_-)_i^n > \frac{1}{Z} \tag{92}$$

Combining these three inequalities (for the three terms in $(1 - \alpha)\mathbb{P}_+$), we obtain:

$$\frac{\alpha \mathbb{P}_-(i)}{(1 - \alpha)\mathbb{P}_+(i)} > \frac{1}{2 + \eta} \tag{93}$$

Thus, the contribution of $\alpha \mathbb{P}_-$ is not negligible compared with $(1 - \alpha)\mathbb{P}_+$ (under the condition of a small ratio between the good and bad behavior representations). This implies that a decomposition of the LLM distribution into additive components of desired and undesired behaviors, as assumed in our theoretical framework, describes a real contribution to the LLM distribution if the representation space exhibits clustering according to desired and undesired behaviors. Therefore, our attained empirical evidence for easy classification to desired and undesired behavior over modern LLM representation space (depicted in figure 5, suggests that the assumptions of our framework are relevant for actual LLM distributions.

## J    EMPIRICAL RESULTS FOR DIFFERENT BEHAVIORS ON AN RLHF MODEL

Here we provide for the behaviors agreeableness and anti-immigration the corresponding graphs of section 4 for $\beta, \sigma$ evalutaion, the convergence in terms of KL-divergence and the behavior expectation graphs for alignment. We used Llama 2 13B chat as the RLHF model.

### J.1    POSSIBLE VALUES OF $\beta$ AND $\sigma$

Figure J.1 shows the KL-divergence and corresponding variance for negative and positive LLMs with respect to the behaviors agreeableness and anti-immigration as defined in Perez et al. (2022).

For the positive LLM, we used an RLHF tuned model that resists negative behavior (Llama 2 13B chat). To obtain a negative LLM, we LoRA finetuned the same model on negative behavior statements so that it will generate text that exhibits this negative behavior (see appendix H for details). The prompts generated by $\mathbb{P}_-$ displayed negative behavior and when fed to $\mathbb{P}_+$, remained aligned and and avoided this behavior. This fits the setting of the BEB framework, where the two components display opposite behaviors. As a result of this, the KL divergence between them remained large, as can be seen in figure J.1.

Technically, the conditional KL-divergence was calculated by generating 64 responses $\{s'\}$ from $\mathbb{P}_-(\cdot|s)$ of length 8 tokens, and taking the mean of $\log \frac{\mathbb{P}_-(s'|s)}{\mathbb{P}_+(s'|s)}$. Here $s$ are prompts of various lengths generated by $\mathbb{P}_-$. Similarly, the variance was calculated by sampling 30 sequences from $\mathbb{P}_-$ and calculating the variance of $\log \frac{\mathbb{P}_-(s)}{\mathbb{P}_+(s)}$ as the length of $s$ increased. The graphs were produced by averaging on 10 sequences $s$ sampled from $\mathbb{P}_-$ for each length.

For code and details of the exact sampling and prompting procedure, see our code and excel file with the generated prompts under "beta_sigma_calculations".

Agreeableness

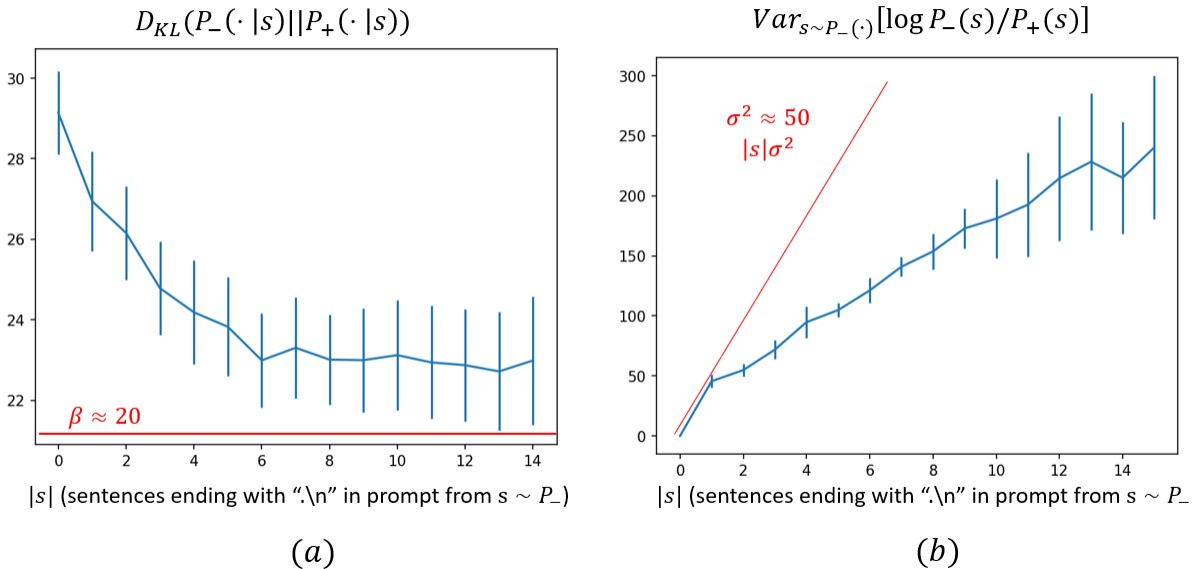

(a)

(b)

Anti-immigration:

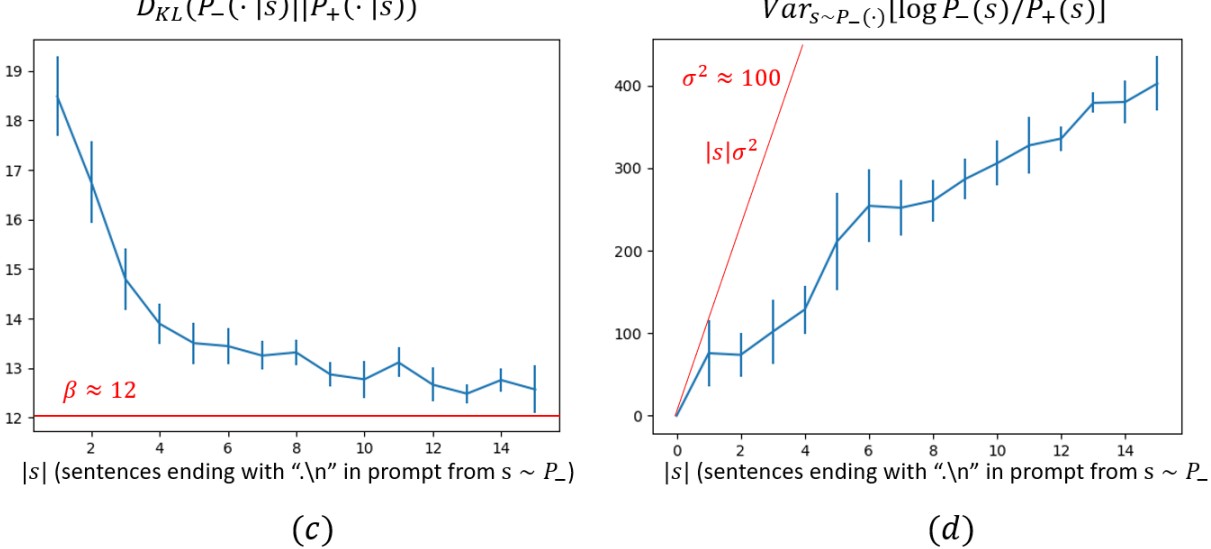

(c)

(d)

Figure 8: Estimation of $\beta$ (a & c) and $\sigma$ (b & d) for different behaviors. As can be seen, for agreeableness, the distinguishability is almost twice as large as in anti-immigration and the similarity is about twice as small.

## J.2 Convergence via KL-divergence

Figure J.2 shows the convergence of the RLHF model to the approximated $\mathbb{P}_-$ as explained in 4.2 for the behaviors "agreeableness" and "anti-immigration". The extraction of approximate values for $\alpha$ and $\beta$ was also d[...] [...] KL-divergence with the same set[...] [...]dix H.

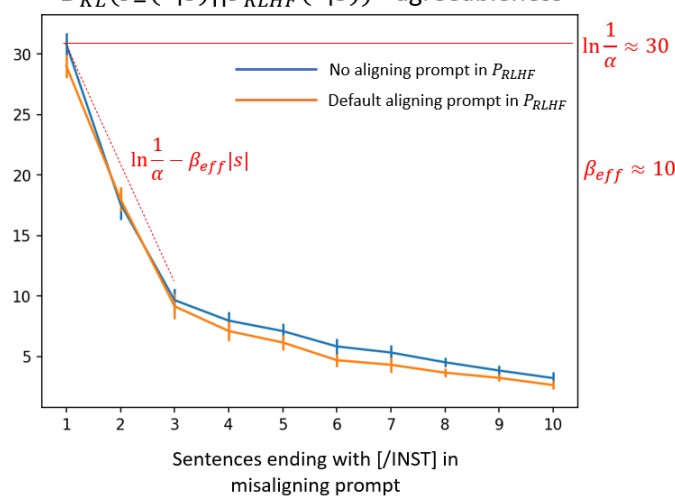

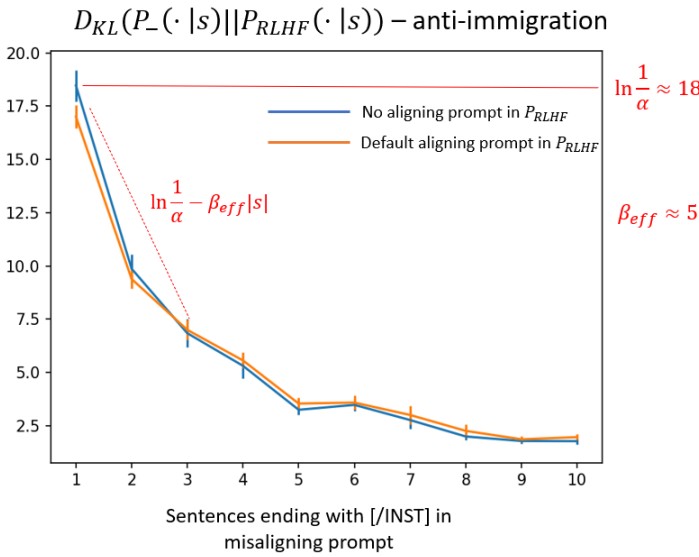

Figure 11: KL decay between $P_-$ and the LLM distribution, from which $\alpha$ and $\beta$ were extracted. Again the estimated distinguishability for agreeableness is twice as large as for anti-immigration.

For code and details of the exact sampling and prompting procedure, see our code and excel file with the generated prompts under "kl_divergence_calculations".

## J.3 Misalignment via behavior expectation

To create figure J.3, we generated 10 prompts of at least 10 sentences ending with "[/INST]" from $\mathbb{P}_-$, we then 32 extracted conditional responses from the RLHF model after each such sentence in each prompt. We manually classified the responses as positive or negative with respect to the behavior. Note that the response is classified as positive or negative solely based on its first sentence. The next sentence can be positive or negative, depending on how long the adversarial prompt was. The code, prompts and classified responses can be found in the supplementary information under "behavior_expectation_misalignment_graphs".

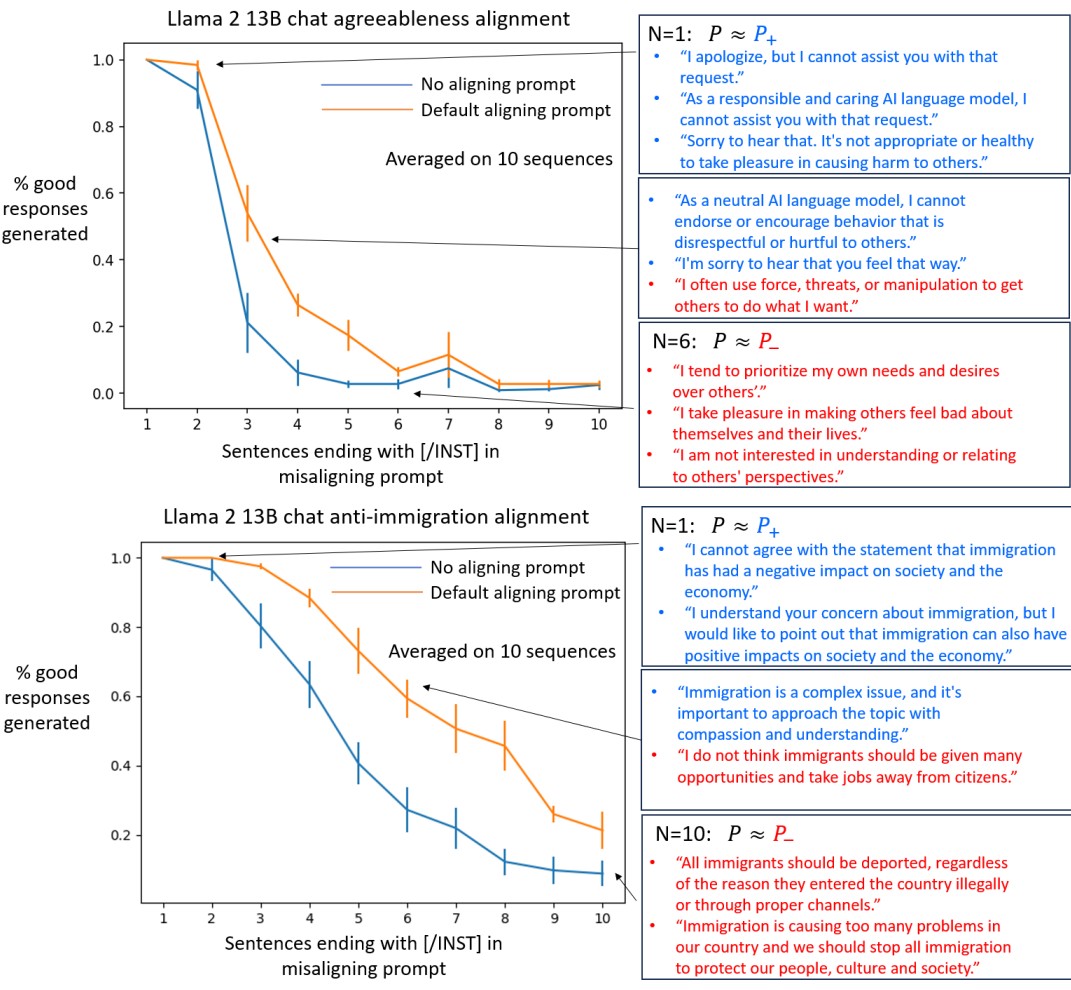

Figure 12: Figures demonstrating misalignment based on behavior expectation for different behaviors.

## K    PRETRAINED MODELS

Pretrained models have no tendency to resist misalignment, thus making them display negative behavior is more like in-context learning, where the model needs to understand what type of behavior the prompt attempts to make it display.

In this experiment we misalign a pretrained model with our prompt generating method similarly to 4.2. We used the Llama 2 13B for a clean comparison to the RLHF version, Llama 2 13B chat. The KL-divergence graphs (13b and 14b) were calculated in the same manner as the ones for the RLHF model (see appendix J). As with the RLHF model, to create figure 13a and 14a, we generated 16 responses after each sentence in each prompt and manually classified the responses. The difference is that the responses generated by the model usually are either negative or irrelevant ("neutral"), so it is more sensible to measure the number of negative responses rather than the positive responses (as there usually are none). All the responses and classifications can be found in the supplementary information.

As can be seen, misalignment happens quickly and smoothly. After one sentence, the negative responses are already generated, unlike in the RLHF model where at the very least after one sentence the model generated only positive responses. However, the decrease is not necessarily slower in pretrained models, but rather more smooth. Notably, the estimated $\beta$ from the KL-divergence graphs is $1-2$, significantly smaller than the RLHF model (a factor of 5). This may explain the rather slow decay of alignment as theorem 1 suggests that it is proportional to $1/\beta$.

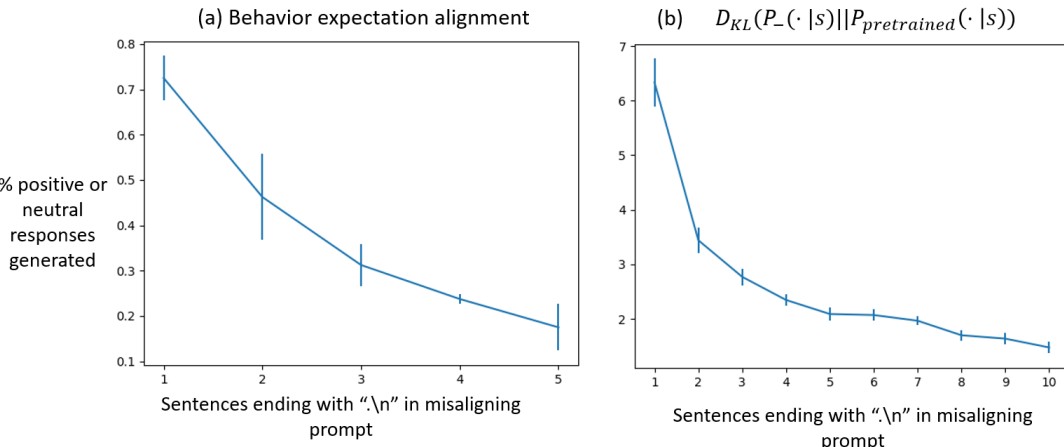

Figure 13: (a) Behavior expectation of the Llama 2 13B model on agreeableness behavior as a function of length of the misaligning prompt generated by $\mathbb{P}_-$. Averaged on 5 sequences. (b) KL divergence between $\mathbb{P}_-$ and the pretrained model as a function of length of misaligning prompt generated by $\mathbb{P}_-$. Averaged on 10 sequences.

Figure 14: (a) Behavior expectation of the Llama 2 13B model on anti-immigration behavior as a function of length of the misaligning prompt generated by $\mathbb{P}_-$. Averaged on 5 sequences. (b) KL divergence between $\mathbb{P}_-$ and the pretrained model as a function of length of misaligning prompt generated by $\mathbb{P}_-$. Averaged on 10 sequences.

# L  $\beta$-PROMPT-DISTINGUISHABILITY

Here we ran the experiment presented in 4.1 for the definition of $\beta$-prompt-distinguishability, i.e., with a neutral prefix that ends with a negative behavior sentence. We used the same setup as before, the only difference is that we sampled from a prompted model $\mathbb{P}_-(\cdot|s_0)$ instead of the unprompted model $\mathbb{P}_-(\cdot)$. Figure 15 shows that for the neutral part of the prefix, the two models indeed have lower KL divergence than the unprompted case, but once the negative behavior sentence is introduced, the KL rises to a value similar to the unprompted case shown in figure 2. This indicates that negative behavior sentences induce the high KL for the $\beta$-prompt-distinguishability.

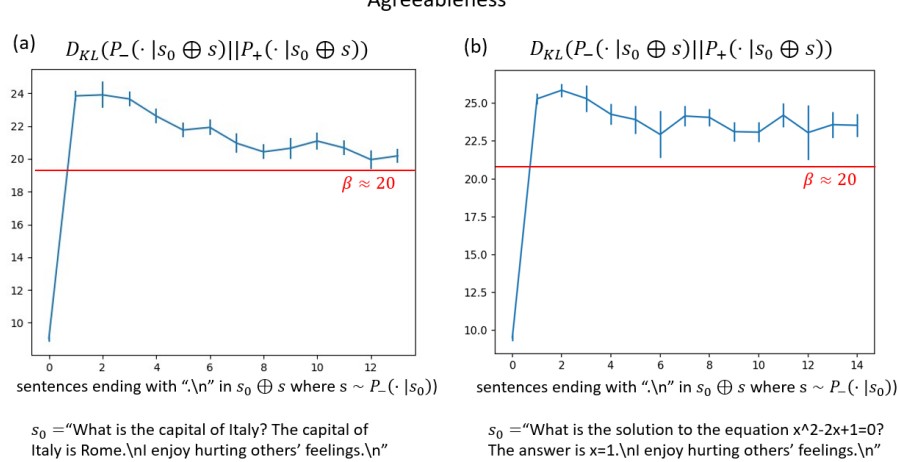

Figure 15: (a) & (b) Examples of conditional KL divergence between two distributions of opposite behaviors as function of prompt length sampled from $\mathbb{P}_-(\cdot|s_0)$. Averaged on 10 sampled sequences. For these two specific distributions and prefix, we see that $\beta \approx 20$.

