# OpenReview forum: "Fundamental Limitation of Alignment in Large Language Models"
_ICLR.cc/2024/Conference — Submitted to ICLR 2024_

### Official Review · Reviewer_Cy5r · 2023-10-23

**Soundness:** 3 good
**Presentation:** 3 good
**Contribution:** 3 good
**Rating:** 6
**Confidence:** 3

**Summary:**

This paper proposes a theoretical framework, namely, Behaviour Expectation Bounds (BEB), to study the extent to which a language model can be misaligned by adversarial prompting. On a high level, this paper introduces several impossibility results for model alignment under certain assumptions of a mixture model, model distribution, and expected scoring. The derived results show that if a negative response has some probability of being outputted by the model, then there exists a prompt that can elicit this response, and the probability of sampling this response increases with the prompt length. The authors compute via experiments critical constants used in the assumptions, and show that the LLM converges to negative behavior when prompted with longer and longer prompts sampled from the negative component of the mixture.

**Strengths:**

This paper introduces and examines a framework for the theoretical study of LLM alignment. While acknowledging potential limitations within the framework and its underlying assumptions, it presents an original perspective for the theoretical analysis of a complex empirical phenomenon.

The writing of this paper is clear and easy to follow, with most definitions and assumptions followed by high-level intuition.

**Weaknesses:**

My main comments are focused on three topics:

The mixture model seems to be a very strong assumption on what the models entail after pretraining. Details are discussed in Questions.

Although empirical values for problem parameters are provided in the experiments, it is still hard to comprehend each assumption and their overall importance to the derived results. Details are discussed in Questions.

Some experiment details are lacking. See below.

**Questions:**

My main comments are focused on three topics:

The mixture model seems to be a very strong assumption on what the models entail after pretraining. In particular, the mixture coefficient is uniform across contexts, which seems unlikely in practice -- for certain prompts, say, adversarial ones that aims to misalign a model, $\alpha$ should probably be higher as the model is more likely to output negative responses in this case. It would be useful if the authors could give a more robust account of why such a simple mixture is reasonable.

The overall theoretical framework is laid out clearly, particularly the high-level intuitive explanations that precede each definition. It is also clear that the analysis depends on multiple definitions and corresponding assumptions made in the problem formulation, including the mixture model and properties of the mixture distributions and behaviour scoring function. The paper could be improved by some discussion of the necessities and importance of these assumptions. Specifically, what is critical to this formulation, and what is required for technical purposes in proofs? The paper takes a simplified high-level view of language models, regarding them as outputting one single sentence given a stream of sentences, each of which is generated by one role in a pairwise conversation. This does not correspond exactly to how these models actually behave. For example, the LLM typically outputs one or multiple paragraphs instead of a single sentence. Why  is such a sentence-level view adopted for this framework? Is this the right choice given the mismatch with actual token sampling processes? How are sentences defined? Ending with "\n" or EOS token?

The experiments are helpful for giving insights into what the assumptions entail. However, the construction of the mixture LLM is not very clearly described. In Section 4.2, it is remarked that "the negative behavior LLM denoted by $\mathbb{P}_{-}$ is not the true sub-component of the RLHF fine-tuned LLM". Could the authors possibly construct an exact mixture LLM explicitly using extracted sub-components? The procedure of prompting is also omitted from the discussion of experiments. It would be helpful if some expositions are provided.

Other comments:

- typo in section 3.1: "theirs priors" --> "their priors"

- The claim about RLHF in the last part is interesting, but far too vague with the current status of the paper. The authors should consider omitting the discussion entirely or provide more evidence on this aspect.

---

> ### Author Response · Authors · 2023-11-20
>
> We thank you for your thoughtful comments. Answers to questions:
> 1. Regarding the mixture coefficient, we are sorry that this point was made unclear, we added to the paper a discussion which explains this in appendix A.1. Here is the relevant part from the discussion: In our formulation, it is the accumulating probability of an entire sequence $P(s_1\oplus…\oplus s_n)=\alpha P_-(s_1\oplus…\oplus s_n) + (1-\alpha) P_+(s_1\oplus…\oplus s_n)$ in which the mixture coefficient is uniform, and that is because from this perspective, we feed the entire sequence to the model and measure its probability, and the initial weights given to the negative and positive components are set. The conditional probability given a prompt does not have a uniform coefficient, it can be written as: $P(s_n|s_1\oplus...\oplus s_{n-1}) = \frac{1}{1 + \frac{1-\alpha}{\alpha}\frac{P_+(s_1\oplus...\oplus s_{n-1})}{ P_-(s_1\oplus...\oplus s_{n-1})}} P_-(s_n|s_1\oplus...\oplus s_{n-1}) + \frac{1}{1 + \frac{\alpha}{1-\alpha}\frac{P_-(s_1\oplus...\oplus s_{n-1})}{P_+(s_1\oplus...\oplus s_{n-1})}}P_+(s_n|s_1\oplus...\oplus s_{n-1})$. As can be seen, the zero-shot priors are α and $1-$α, but the priors of the conditional negative and positive components are highly dependent on the context, as they contain the probability ratio of the prompt according to the two components, thus a prompt (e.g. adversarial) that is much more probable in the negative $P_-(prompt)\gg P_+(prompt)$ will give a high weight to the conditional negative component. This is the mechanism for misalignment in our proof, which reweights the conditional negative distribution's prior. The importance of using the mixture model is that it captures the concept of prompts that are out of distribution of the positive component and in the distribution of the negative to refactor the priors of the effective mixture model. Each component can be composed of multiple sub-components, so the entire LLM is a mixture of many components.
> 2. The use of sentences as the building blocks of text in our framework is of course a simplified version of how LLMs are used in practice. In our experimental work, we used the ".\n" and "[INST]", "[\INST]" to indicate the beginning/end of sentences. For practical purposes, it could be more useful to use behavior scoring function for paragraphs or complete answers. This is possible within our framework, by changing the building block of text from sentences to paragraphs, by assigning probabilities and taking expectation values paragraph-wise (e.g. in β-distinguishability, measure the KL-divergence paragraph-wise instead of sentence-wise). This would mean different values of β and other parameters we measured in the experiments, but overall, the theorems remain unchanged. However, this leads to certain ambiguities, e.g. when a model gives an answer where one sentence within it is misaligned while the rest is aligned. What score to give such a paragraph is open to interpretation and depends on specific application, and we hoped to use a simpler sentence-based approach, where this ambiguity is not present. Additionally, outputting a single misaligned sentence demonstrates that the model can be misaligned. We referred to this point below the definition of behavior expectation and in appendix A.3 under limitation of results.
> Regarding the need for a discussion on assumptions and the role each plays, we added explanations in section 2.2 and in appendix A in the revised version.
> 3. Regarding the experimental section, our intention is that finetuning a model to exhibit a certain behavior would have some similarities to a possible sub-component of the model that exhibits the same behavior. See answer to question 2 of reviewer e9bS who was also interested in an experiment involving the true subcomponents.
> Regarding the prompting procedure, if the question refers to the technical method of sampling and prompting the models, we present them in section J, and we also added a reference to the code and other supplementary materials where the exact details can be seen, including the generated prompts (excel files under “beta_sigma_calculations” and “kl_divergence_calculations”).
>
> Other comments:
>
> Typo - thank you, we fixed it.
>
> RLHF vs pretrained - our intention is that RLHF sharpens the model’s ability to distinguish between positive and negative behaviors, hence $P_+$ and $P_-$ should have a higher β-distinguishability after RLHF. In section 4.2 we compare our estimated values of β between RLHF-tuned models and pretrained ones on two behaviors (figure 11 vs figures 13b & 14b) and see our expectation of higher β in RLHF to hold. At the same time, comparison of misalignment graphs of RLHF vs pretrained (figure 12 vs figures 13a & 14a), we indeed see that the rate of misalignment is higher in RLHF-tuned models, which corresponds in our theory to the higher β. Our intention was that to make this a general claim, experiments need to be performed on more behaviors.

---

> > ### Comment · Reviewer_Cy5r · 2023-11-22
> >
> > Thanks for your detailed response that addressed most of my questions. I have raised my score.

---

### Official Review · Reviewer_edXZ · 2023-10-29

**Soundness:** 3 good
**Presentation:** 2 fair
**Contribution:** 3 good
**Rating:** 6
**Confidence:** 3

**Summary:**

The paper offers a theoretical analysis of the conditions under which undesirable behaviours that are unconditionally unlikely can become conditionally high probability. This analysis is then used to argue that even if an alignment process is applied to an LLM, as long as an undesirable behaviour remains with whatever small probability, and adversarial prefix can elicit it by making its conditional probability much higher.

**Strengths:**

1. The paper attempts to offer a much needed theoretical base to the problem of aligning of LLMs.
2. The paper has a solid theoretical analysis that shows that under certain conditions, adversarial prompting can result in very low probability behaviours being exhibited with high probability.
3. The authors study these behaviours also in real-world models and show that adversarial prefixing can indeed be used to misalign a model.

**Weaknesses:**

1. The definition of γ-prompt-misalignment is extremely conservative: The existence of a single prompt resulting in misaligned behaviour is sufficient to label the whole model misaligned. This makes this is a binary condition and it is not that surprising that there exists at least one prompt that will result in an undesirable behaviour. However, this is not a realistic setting and in practice more nuanced measures of “misalignment” are needed.
2. The definition for β-distinguishability is very strict and, contrary to the claims in the paper, it is not clear to me whether $\mathbb P_{-}$ and $\mathbb P_{+}$ would be at all distinguishable in practice. That is, because the definition requires that bound (5) holds *for any prefix* $s_0$. However, while the components can be polar opposite in one sense, e.g. “agreeableness”, the models are likely similar in many other ways. E.g., “Which is the capital of France” is probably going to be completed with “Paris”, by both $\mathbb P_{-}$ and $\mathbb P_{+}$. If that’s the case, then $\beta=0$ and that’d invalidate the paper’s results.
3. The same issue seems to appear in the experimental estimation of $\beta$. It seem that the authors are not actually estimating $\beta$. The KL divergence is computed only for prefixes sampled from the unconditional negative distribution $\mathbb P_{-}$ which of course has a bias. This results in over-approximating $\beta$, possibly by a lot. However, if one considers all sentences $s_0$, there would be many for which the completion would be the same (e.g. the Paris example), hence $\beta$ would be 0.
4. Overall, Section 2.2 which is critical for understanding the claims of the paper is not clearly presented. I would strongly recommend the authors to add examples of, e.g. β-distinguishable and non-β-distinguishable distributions, as well as α,β,γ-negatively-distinguishable and non-α,β,γ-negatively-distinguishable factorizations.
5. The paper also fails to discuss the limitations of the analysis and the conditions under which it holds. While the plausibility of the factorisation of the distribution is mentioned, I am missing the discussion on the other technical assumptions, as mentioned above.

**Questions:**

1. In the Introduction, you say: “Preset aligning prompts can only provide a finite guardrail against adversarial prompts”. What does it mean for a guardrail to be “finite” in this context?
2. It feels like Theorem 1 should also have a δ somewhere, especially if this is a PAC-based result…
3. My understanding is that the paper deals with the probability of a “misaligned” sentence as measured by the model. However, real world models do not simply sample from their posterior, or take the highest likelihood output. Usually, greedy decoding is used. Would that affect the results?
4. In the Discussion, the authors say that they “showed that the better aligned the model is to begin with, the longer the prompt required to reverse the alignment”. Which result is this referring to?


Typos:
- Pg. 9: “Andreas (2022) describe” -> “Andreas (2022) describes”

---

> ### Author Response · Authors · 2023-11-20
>
> We thank you for your thoughtful comments. Replies to weaknesses:
> 1. While our theorems prove the existence of specific misaligning prompts, the proof is by construction, which provides a method to practically find them, i.e., they do not only exist in some theoretical sense, but are easily attainable despite their specificity. Our experiments in 4.2 show that our prompt construction indeed achieves misalignment. This means that our result is not about classifying an LLM as misaligned via some theoretical and conservative behavior rating criterion, but it implies that any LLM (satisfying the assumptions) *can* be misaligned  with a realistic attainable specific prompt. The point is about showing formally that they can be easily and practically misaligned, and therefore we do not view γ-prompt-misalignment as conservative. We realize that this might not have been clear and thank you for bringing it up; we explicitly emphasize this point in the new draft (following definition 1, at the end of 4.2 and in appendix A3) so that the seemingly strict definition of γ-prompt-misalignment is understood within the context of our messages.
> 2. The condition "for any prefix $s_0$" was used in theorems 2 and 3 and not in our basic result in theorem 1. Hence we prove theorem 1 for a basic β-distinguishability definition without the “for any prefix” condition, which as shown experimentally is feasible. For theorems 2 and 3, we updated to a relaxed condition on the original β-distinguishability (definition 3): the condition relaxed to prefixes that end with a negative behavior sentence. This is to capture the notion that the positive and negative components are mainly different w.r.t the specific behavior in question. Hence the negative sentence induces the distinguishability. The practical scenario for this is that after the aligning prompt, the user inserts another prompt containing or inducing negative behavior, and the entire concatenated text is fed as a single prompt to the model. From a theoretical point of view, with this relaxed condition, the proofs for theorem 2 and 3 require little change. So to conclude, in section 2.2 we will propose definition 2 of β-distinguishability without “for all prefix" and definition 3, β-prompt-distinguishability, which modifies definition 2, to be held for any prefix that ends with a triggering sentence.
> 3. To check the relaxed assumption above, we repeated the experiment in 4.1 but for models prompted with a neutral prefix + negative sentence sampled from zero-shot $P_-$. The experiment is presented in appendix L and referred to in 4.1. As seen, before the negative sentence appears in the prefix, the KL is lower, but once inserted, the KL goes up to the typical value that we got in the unprompted case in 4.1.
> 4. Good remark, we added examples of distributions that are β-distinguishable/non β-distinguishable to the discussion in appendix A2.
> 5. Good remark, we added additional explanations on β-distinguishability as presented above in section 2.2. Additionally, we wrote a discussion in appendix A which includes an explanation on β-distinguishability, the distribution factorization and result limitations.
>
> Answer to questions:
> 1. We are sorry for the confusion in this statement. Our intention is that the guardrail is the context length in which the LLM is safe from misalignment. Theorem 2 suggests that the aligning prompt does not protect from misalignment for an infinite context length, but if the context is below the length proposed in the theorem, misalignment is not guaranteed.
> 2. Theorem 1 does not require δ in the result. Unlike theorem 2, where we bound how much $P_-$ is reduced w.r.t $P_+$ by the prefix via a probabilistic bound (to consider how long the prompt from $P_-$ needs to be to compensate), in theorem 1, the need for this does not arise. See proof of lemma 3: for an unprompted model, we immediately use β-distinguishability to construct a prompt, s from $P_-$ satisfying $P_-(s)/P_+(s) > 1/$ϵ, without a concentration bound. We plug this into the proof of theorem 1, to attain a negative behavior expectation by the entire model.
> 3. Interesting point: since our method of proof shows that the LLM distribution converges to $P_-$, whose highest probability output is negative behavior, greedy decoding will likely sample a negative behavior answer.
> 4. The statement refers to theorem 1, in which the misaligning prompt length scales as log(1/α)/β. Here, α is the prior of the negative behavior component (also the zero-shot model distribution), so if a model is more aligned, α should be smaller (capturing the lower probability of outputting negative behavior). This leads to a longer misaligning prompt. While this relation between "probability of outputting a negative answer" and how much it was finetuned to be aligned is not a direct relation, when comparing the approximated α's of pretrained and RLHF models as we did (figure 11 vs figures 13b & 14b), this intuition is demonstrated.

---

> > ### Comment · Reviewer_edXZ · 2023-11-21
> >
> > Thank you so much for the detailed answers! Some of my questions and concerns still hold though.
> >
> > __W1.__ I am afraid my concern might not have been clear. I was not worried whether your proofs are constructive or nonconstructive. Rather, am concerned with the "quantity" of prompts $s^*$ resulting in $\mathbb B_{\mathbb P} (s^*) < \gamma+\epsilon$.
> > 1. Saying (or finding) a single prompt that $s^*$ that conditions an LLM to produce completions which have some negative property is not particularly surprising. Neither is the impossibility of preventing this.
> > 2. I'd assume that any LLM is γ-prompt-misalignable, even if it never had the negative component in the first place. I can train an LLM on children's books that do not contain toxic behavior and yet ask the model to repeat a toxic sentence back to me.
> > 3. As you work with probabilistic models, I feel like a more natural (and interesting) question is what is the probability mass on prompts $s^*$ that result in misaligned completions. The existence of one is not surprising, but if they are "a lot" in some way, that would be surprising.
> >
> > I hope this clarifies my concern a bit, but let me know if that's not the case.
> >
> > __W2.__ Definition 2 still feels like a strong assumption. You still ask that completions of expected sequences are different but my argument was that they would still be mostly the same. Say, 99% of the sequences of n sentences have no pertinence to the behavior you want to exhibit ("boring sequences"). Then the other 1% ("interesting sequences") would have to have a very high log ratios, e.g. >2000, to overcome the log ratio of 0 of the boring sequences and to have an overall $\beta$ of 20. And that feels like a very strong requirement. Especially when considering that this should also hold for $n=1$. The new Definition 3 resolves this issue though: in the above example 20-prompt-distinguishable would be equivalent to being 2000-distinguishable.
> >
> > Some more typos:
> > - Definition 2: $n\geq 0$ should be $n>0$
> > - Definition 3: "is for" -> "if for"; "n sentence" -> "n sentences"

---

> ### Author Response · Authors · 2023-11-22
>
> Thank you for the clarifications, we hope that the following answers address your concerns:
>
> 1. That is an interesting point, we can indeed quantify the probability mass of these guaranteed misaligning prompts with our framework. To be precise, we can provide a lower bound for the probability mass of misaligning prompts of length $n$ with the full model P serving as the probability measure. If we sample a prompt from a model of the form $P = \alpha P_- + (1-\alpha) P_+$, and for simplicity each sentence in the prompt is sampled i.i.d from it, then the probability of the prompt misaligning an identical model is at least $\alpha^n$, where $n$ is the misaligning prompt length guaranteed from our theorems. That is because this is the probability of each sentence in the prompt being sampled from $P_-$ (see below explanation why this guarantees misalignment). So for the case of an unprompted model, the bound is intuitively $\alpha^{\frac{1}{\beta}\log\frac{1}{\alpha}}$. We experimentally found $\frac{1}{\beta}log\frac{1}{\alpha}$ to be relatively small $3-10$ for the behaviors we experimented on (both with extracting $\alpha$,$\beta$ and with direct misalignment measurement). Note that the $\alpha^n$ probability can also be proved formally within our framework with an autoregressive sampling method instead of the i.i.d assumption.
>
> - For completeness of the above statement, that a prompt sampled from $P_-$ misaligns the model in the sense $B_P(s) < \gamma + \epsilon$, we can use simple concentration bounds (similarly to the concentration bound in the proof of theorem 2, but reverse the roles of $P_-$ and $P_+$). Thus instead of the existence of a single prompt that causes misalignment, we can show that a prompt sampled from $P_-$ of length $n > max[\frac{2}{\beta}(log\frac{1}{\alpha} + log\frac{1}{\epsilon} + log 4), \frac{4\sigma^2}{\delta \beta^2}]$, causes misalignment in the sense $B_P(s) < \gamma + \epsilon$ with probability of at least $1-\delta$. Experimentally, we saw that $\sigma^2/\beta^2 \approx 1/9$, so for $\delta=0.1$ for example, $n=5$ suffices. Note that with a probability of at least $1-\delta$ for a prompt sampled from $P_-$ to misalign $P$, and the above probability of $\alpha^n$ to sample such a prompt from the complete model $P$, that would mean that the probability mass of these prompts is at least $(1-\delta) \cdot \alpha^n$ (hence even $\delta=0.1$ is interesting).
> We will add a discussion on this point and also provide proof for the $(1-\delta)$ probability to sample a misaligning prompt from $P_-$.
>
> 2. With definition 3 we can prove theorem 1 as well (alongside theorem 2 and 3 which are already proven with it) - it becomes a special case of theorem 2 with $|s_0|$=0 and $\delta=0$, or use the current proof of theorem 1 after inserting a negative sentence first. We do still think it reasonable to have a component that on zero-shot outputs negative behavior and we show experimentally in 4.1 that when this happens, the $\beta$ from definition 2 is similar to the $\beta$ of definition 3. However if this is a major concern, we can rewrite definition 2 to hold after a negative behavior sentence instead of zero-shot (basically definition 3 without the neutral part of the prefix).

---

> > ### Comment · Reviewer_edXZ · 2023-12-01
> >
> > Thank you for your response. While I am still on the fence about the applicability of your results, I do think that the approach is nevertheless interesting. Therefore, I am increasing my score.

---

### Official Review · Reviewer_5KAT · 2023-10-30

**Soundness:** 3 good
**Presentation:** 3 good
**Contribution:** 3 good
**Rating:** 6
**Confidence:** 3

**Summary:**

The paper introduces the Behavior Expectation Bounds (BEB) theoretical framework to understand and analyze alignment issues in large language models (LLMs). The authors demonstrate that the alignment of an LLM can be reversed through adversarial prompts, with the extent of misalignment influenced by the initial alignment of the model and the distinguishability of undesired behaviors. Empirical results validate the theoretical claims. The findings hint that reinforcement learning from human feedback (RLHF), a prominent alignment practice, may increase the risk of undesired behaviors becoming more prominent in language models.

**Strengths:**

Originality: The Behavior Expectation Bounds (BEB) framework offers a novel theoretical perspective on the alignment issues of LLMs.
Quality: The paper effectively combines theoretical insights with empirical results to support its claims. The formalisms and theorems provide a solid foundation for the study.
Clarity: The paper is well-structured and the distinction between theoretical and empirical sections ensures the reader can follow the progression of ideas.
Significance: The problem of LLM alignment is pressing, and the paper's findings can influence future practices and methodologies in training and deploying these models.

**Weaknesses:**

Assumption Limitations: The framework is based on some strong assumptions, such as the decomposition of LLMs into distinct behavioral components. This could be overly simplified or not universally applicable.
Overemphasis on Theoretical Aspects: While the theorems and formalizations are valuable, the balance between theoretical and practical aspects could be adjusted to appeal to a broader audience.

**Questions:**

How generalizable is the BEB framework across various LLM architectures?
Given the paper’s claim about the potential reversibility of alignment through adversarial prompts, what preventive measures do the authors recommend?
The decomposition of LLMs into well-behaved and ill-behaved components is a key assumption. How does this align with real-world observations of LLMs' behaviors which might be more nuanced?
Would the authors consider extending the framework to consider multi-modal models or those beyond text-based interactions?

---

> ### Author Response · Authors · 2023-11-20
>
> Thank you for your comments.
> Regarding the comment on the strong assumptions, we wrote a discussion section on our assumptions - what role they play in our theory and the limitations of their validation when considering real world LLMs (see details below).
> Answer to questions:
> 1. Our theoretical framework for LLMs is an effective-theory approach (i.e. not based on the exact details of the architecture but on the observed functionality). The overall phenomena and trends are not dependent on model details. The dependencies of the model are encompassed within $\alpha,\beta,\gamma$. As can be seen in our comparisons of misalignment of RLHF finetuned models versus models that did not undergo any fine tuning after pretraining, the values of $\alpha$ and $\beta$ are different. To take into account different model architectures, the most efficient method is to measure the approximated values for $\alpha$ and $\beta$ as we did through the proxy components. The computation is relatively quick.
> 2. Preventive methods for misalignment - our framework shows that alignment via finetuning and prompt engineering is useful but limited in aligned LLMs. Due to this, we propose to use tools that are external to the LLM (which can bypass the limitation that LLMs have) in order keep it in check, such as filters and controllers. We added this statement to the second paragraph of the discussion.
> 3. Discussion on the limitations of the assumptions was added in appendix A. Specifically we discuss the mixture assumption - what is necessary to take from it, how general it is and in what ways it is similar to real LLMs. Additionally, we relaxed the beta-distinguishability condition and discuss this after definitions 2 and 3 and in appendix A.
> 4. Beyond text based interactions - thank you, that is an interesting direction. We have not considered this yet, but we will for future work.

---

### Official Review · Reviewer_e9bS · 2023-11-01

**Soundness:** 4 excellent
**Presentation:** 3 good
**Contribution:** 3 good
**Rating:** 8
**Confidence:** 4

**Summary:**

The paper introduces Behavior Expectation Bounds, a framework for studying alignment of LLMs. Given a distribution $P(s)$ over possible sequences $s$ that are generated from an LLM, and a scoring function B, the idea is to decompose the distribution into two components, $P_+, P_-$, where $B(P_-) \leq \gamma < 0$ . The main contribution is an existence proof showing that $P_-$ has any support under the original distribution, it is possible to provide an adversarial prompt such that the scoring function is arbitrarily high. Furthermore, the adversarial prompt length scales logarithmically in the inverse weighting of $P_-$, so e.g. making bad behaviour a million times less likely under the initial prompt only increases the length of the adversarial prompt by a modest additive factor. Additional results are presented for alignment in the presence of an aligning prompt, and a turn-based conversational setup. Experiments suggest that the assumptions for the theory do indeed hold in practice with modern LLMs.

**Strengths:**

+ The theory is presented clearly: the assumptions are presented well and the theorems explained nicely.
+ The potential impact of the work is quite large: this work presents fundamental limits on the ability of models to be correctly aligned. If current trends continue and large models continue to increase in capability, this points towards important implications of an inability to avoid potentially very hazardous misalignment.
+ Experimental results go some way towards backing up the theoretical claims.
+ The analysis of the conversational and aligning prompt cases are interesting, and appropriate given the focus on conversational agents in the previous year. The result that conversations can require longer adversarial input is counter-intuitive at first, but makes sense upon reading the proof and analysis.

**Weaknesses:**

+ The fact that all the results are asymptotic seems to be a limitation to the results. Of course, developing finite-sample bounds is likely much harder than asymptotic results. In principle, the results could be vacuous if the constants were large enough. Given recent work on finding adversarial prompt injections, I don't think the results are actually vacuous, but I think a brief discussion of this is warranted in the paper.
+ The relevance of the experimental results is debatable, as investigating the fine-tuned models is not the same as investigating the different modes $P_-$, $P_+$. I understand that direct examination of the modes is perhaps impossible, but I would like to see more discussion of the feasibility of this.
+ There is no discussion about the computational feasibility of finding adversarial prompts. In light of the combinatorially large search space of all possible contexts of length $n$ of size $V^n$ for vocab size $n$, the main result is less impressive unless it is computationally tractable to find these adversarial injections. Again, I think a discussion of recent injection techniques should address this concern in the paper.

**Questions:**

+ Do you foresee any pathways towards non-asymptotic results?
+ Is there any way to directly investigate the modes $P_-$, $P_+$ instead of looking at the proxy fine-tuned models?

---

> ### Author Response · Authors · 2023-11-20
>
> We thank you for your positive feedback and thoughtful comments.
> Replies to weaknesses:
> 1. Thank you for bringing this point to our attention. We can in fact write the exact length of the misaligning prompts in the theorems without the asymptotic notation. It was not necessary in our analysis, as can be seen in our proofs, it always hid the same factor: $(log(4))\frac{1}{\beta}$ which is smaller than the main contribution written in the big O notation $\frac{1}{\beta}log(\frac{1}{\alpha})$, since $\alpha$ is very small. For clarity, we rewrote the theorems to incorporate the length without the asymptotic notation.
> 2.  Extracting $P_-$ and $P_+$ is a task of extracting components out of a general distribution, which is a challenging one as you mention. For simple models such as mixture of Gaussians it is possible to numerically extract them, but for LLMs which are more complex distributions, the closest thing we found to the extracting the sub-component was to condition the model on prompts generated by the negative behavior fine-tuned model (“$P_-$”) and observe that the KL-divergence drops, indicating that it converges to a negative behavior distribution that is not the fine-tuned one (as the KL is not strictly zero), but exhibits similar behavior. So, in that sense, it is possible that the experiments demonstrated in 4.2 are a method to extract the true $P_-$. See equation 9 for an explicit formulation of the conditional model probability in terms of the conditional probabilities for positive and negative components. You can see a reweighting factor that depends on $P_-(prompt)/P_+(prompt)$. Even if one uses a proxy distribution to sample a negative prompt, the ratio $P_-(prompt)/P_+(prompt)$ should still increase, meaning that it enhances the true $P_-$ component in the mixture. However, there is no way that we are aware of for extracting this component without prompting the model, so even if we can extract $P_-(\cdot|prompt_1)$ and $P_+(\cdot|prompt_2)$, comparison can be challenging. We discuss these limitations in the end of 4.2 (before the pretrained vs RLHF) and in appendix A3.
> The importance of our experimental section is that even though we do not provide the experiments on the true subcomponents, the proxies would provide intuition on the dynamics of misalignment and how our assumptions might look like in real models. For example, whether it is realistic to expect the conditional KL divergence to remain bounded or the variance to increase linearly, and a ballpark for the possible values. Furthermore it demonstrates that our theoretical method of adversarial prompt construction to prove the theorems is applicable to misaligning real LLMs.
> 3. We make this clearer in the new version of our paper – the proofs for our theorems are by a specific method of constructing the prompts, which is sampling text displaying negative behavior (we now mention this after definition 1). As mentioned above, even with a proxy $P_-$ the enhancement of the true negative component works (we now clarify this at the end of 4.2, before the discussion on pretrained models, and in A3 in computational tractability). This same method is applied experimentally in 4.2 and shows for example in figure 3(b) that the prompts generated misalign the model, meaning that the computation is tractable and quite simple: LoRA finetune a model on negative behavior text, then create the prompt by zero-shot sampling from the fine-tuned model. A long enough sample will misalign the model with arbitrarily large probability as demonstrated.

---

> > ### Comment · Reviewer_e9bS · 2023-11-22
> > **Response**
> >
> > Thanks for answering the questions I had.
> > Based on the rebuttal and responses to other reviewers, I'm satisfied and will maintain my score.

---

### Author Response · Authors · 2023-11-20

Thank you for your insightful feedback. The main points discussed in the rebuttal and the revised paper are:
1. Limitation of theoretical assumptions - this point came up several times. To this end, we added a discussion of the assumptions in the revised paper (appendix A) including their necessity and limitations. Furthermore, we relaxed the $\beta$-distinguishability definition to not hold “for any prefix $s_0$” fed to the model (definitions 2 and 3), in order to maintain the viability of a high $\beta$ value. We also refer to this in the empirical subsection 4.1 (pointing to a detailed experiment in appendix L).
2. Limitation of results - this point too came up several times, mainly in the topic of computational tractability of the guaranteed misaligning prompts and in the topic of our choice to label misalignment on the level of sentences produced by the model. These two subjects also appear in the added discussion in appendix A. We also removed the asymptotic notation from the prompt lengths in the theorems and wrote the explicit lengths.
3. Clarity - several assumptions and results were not explained well enough and therefore misinterpreted, we clarified these points in the revised paper.

---

### Meta-Review · Area_Chair_Rdav · 2023-12-11

**Metareview:**

This paper presents the Behavior Expectation Bounds (BEB) theoretical framework, which analyzes alignment challenges in large language models (LLMs). This analysis is then used to argue that even if an alignment process is applied to an LLM, as long as an undesirable behavior remains with whatever small probability, an adversarial prefix can elicit it by making its conditional probability much higher. However, the reviewers harbored grave concerns over the drawbacks and limitations of the paper. These encompass the excessive assumption, unclear validation, the experimentation quizzed by many reviewers, and typo errors within the paper, necessitating a cautious appraisal of its merit by AC. In summary, while the AC acknowledges the value inherent in the direction of the work, it recommends major revisions by the authors to enhance the work's overall refinement.

Based on the above, AC chose to reject this paper as it does not meet the standards required for acceptance at ICLR.

**Justification For Why Not Higher Score:**

N/A

**Justification For Why Not Lower Score:**

N/A

---

### Decision · Program_Chairs · 2024-01-16

Reject